# Implication of Mineralogy and Isotope Data on the Origin of the Permian Basic Volcanic Rocks of the Hronicum (Slovakia, Western Carpathians)

Ján Spišiak [1,*], Anna Vozárová [2], Jozef Vozár [3], Štefan Ferenc [1], Viera Šimonová [1] and Juraj Butek [1]

1   Department of Geography and Geology, Faculty of Natural Sciences, Matej Bel University, Tajovského 40, SK-97401 Banská Bystrica, Slovakia; stefan.ferenc@umb.sk (Š.F.); viera.simonova@umb.sk (V.Š.); juraj.butek@umb.sk (J.B.)
2   Department of Mineralogy and Petrology, Faculty of Natural Sciences, Comenius University, Mlynská Dolina, SK-84215 Bratislava, Slovakia; anna.vozarova@uniba.sk
3   Earth Science Institute Slovak Academy of Science, Dúbravská cesta 9, SK-84005 Bratislava, Slovakia; jozef.vozar@savba.sk
*   Correspondence: jan.spisiak@umb.sk

**Abstract:** The paper presents new geochemical data of the basic to intermediate volcanic rocks from the Hronic unit. The studied rocks are strongly altered and the primary mineral composition (clinopyroxenes, amphiboles, spinels, biotites, and plagioclases) is preserved only in some localities. The clinopyroxene corresponds to augite and primary amphiboles to pargasites. Spinels have a chemical composition similar to spinels from volcanic rocks. According to chemical composition, the studied basalts correspond to rift-related continental tholeiites. $^{143}Nd/^{144}Nd$ vs. $^{87}Sr/^{86}Sr$ isotopic ratios of the basalts are scattered around the value of $^{143}Nd/^{144}Nd$ for CHUR, where $^{143}Nd/^{144}Nd$ ratios are relatively stable and $^{87}Sr/^{86}Sr$ values are relatively varied. According to isotopic ratio of $^{207}Pb/^{204}Pb$ vs. $^{206}Pb/^{204}Pb$, the basalt analyses lie in the field of EMII (mantle source enriched with crustal materials). The new results of U-Pb LA-ICP-MS dating of apatite point to an age $254 \pm 23$ Ma (Lopingian).

**Keywords:** Western Carpathians; Hronic Unit; Permian volcanic rocks; geochemistry; Sm/Nd and Pb isotopes

## 1. Introduction

The Pennsylvanian–Permian continental sedimentary basin of the Hronicum represents a type of retroforeland basin [1,2], whose origin has been genetically associated with the convergent regime in the final stages of the Western Carpathian Variscan orogeny. This sedimentary basin is characterized by a huge amount of red-beds type siliciclastic sediments accompanied by basic to intermediate volcanic rocks. Relics of the original filling of this sedimentation area are preserved in today's Western Carpathian structure within a system of superficial nappes, whereas they are completely tectonically detached from their original basement. The crustal character of the original basement was inferred from petrological analysis of sediments [3–5] as well as isotopic ages of clastic mica [6], monazite [7], and zircon [8]. The results documented an active continental margin crust provenance, with dominant Devonian–Mississippian and Pennsylvanian magmatic sources.

This article studied basic to intermediate volcanic rocks whose genesis and chemical composition are closely related to the tectonic processes that led to the formation of the Hronicum Pennsylvanian–Permian sedimentary realm. Previous analyses of the Hronicum basalts/andesites have been focused on their field distribution and description of structural relations and especially on their mineral and chemical composition [9–11]. They were generally classified as rift-related continental tholeiites.

In this paper, we briefly review the main characteristics of the Permian magmatism in the Hronic unit domain and present new chemical and isotopic data in an attempt to provide some insights into its origin and significance. The main goal of this study is to bring new petrological and geochemical data and identify in more detail the sources of magmas and differentiation processes, and thus infer the relationships with the geotectonic setting of the studied area. Therefore, nine additional samples were collected from the Hronicum Permian volcanic rocks for a better temporal covering of the Hronic formation (Table 1). The studied samples were collected from the Permian volcanites of the 1st and 2nd eruption phases as well as from the dykes crossing Pennsylvanian sediments.

**Table 1.** Location of samples.

| Sample No. | Rock Type | Sample Locality | GPS Coordinates |
|---|---|---|---|
| M-1 | coarse-grained gabbro-diorite dyke | Malužinská dolina Valley, cliff on the right slope of the forest road 812 m a.s.l. | 48°57.182′ N 19°48.783′ E |
| M-1A | fine-grained gabbro-diorite dyke | Malužinská dolina Valley, cliff on the right slope of the forest road 812 m a.s.l. | 48°57.182′ N 19°48.783′ E |
| M-2 | andesite/basalt-2nd eruption phase | Hodruša Valley, SE from the Malužiná Village, cliff on the right slope of the forest road, 870m a.s.l. | 48°57.552′ N 19°50.051′ E |
| M-3 | andesite/basalt-2nd eruption phase | Malužinská dolina Valley, forest road cut, 1.4 km SE from Malužiná Village, 770 m a.s.l. | 48°57.544′ N 19°50.091′ E |
| M-4 | andesite/basalt-1st eruption phase | N from the Liptovská Teplička Village, abandoned quarry along the Šuňava-Liptovská Teplička road, 890 m a.s.l. | 48°59.038′ N 20°05.353′ E |
| M-5 | basaltic andesite-1st eruption phase | Benkovský potok Valley, small quarry at the left side of the forest road, 832 m a.s.l. | 49°00.468′ N 19°59.747′ E |
| M-7 | basaltic andesite-1st eruption phase | S from the Hranovnica Village, forest road at the left slope of the Vernár-Hranovnica state road, 694 m a.s.l. | 48°57.970′ N 20°17.566′ E |
| M-8 | coarse-grained gabbro-diorite dyke | West end of the Nižná Boca Village, 860 m a.s.l. | 48°56.915′ N 19°46.157′ E |
| M-9 | andesite/basalt-2nd eruption phase | Kvetnica quarry, 725 m a.s.l. | 49°00.521′ N 20°17.085′ E |

## 2. Geological Background

The Hronic nappe system (Figure 1) structurally represents the highest tectonic unit in the Central Western Carpathians. There is no generally accepted opinion about the structure and composition of the Hronic nappe system and its palaeogeographic position. A lot of theories have emerged in the past four decades of research, with current definition of Hronicum as a rootless superficial nappe system [12,13] that mostly comprises a Carboniferous–Permian volcano-sedimentary sequence and Triassic carbonate sediments (with significant similarities of the latter to the Oberostalpine-type). Jurassic-Lower Cretaceous sediments are preserved only locally [14].

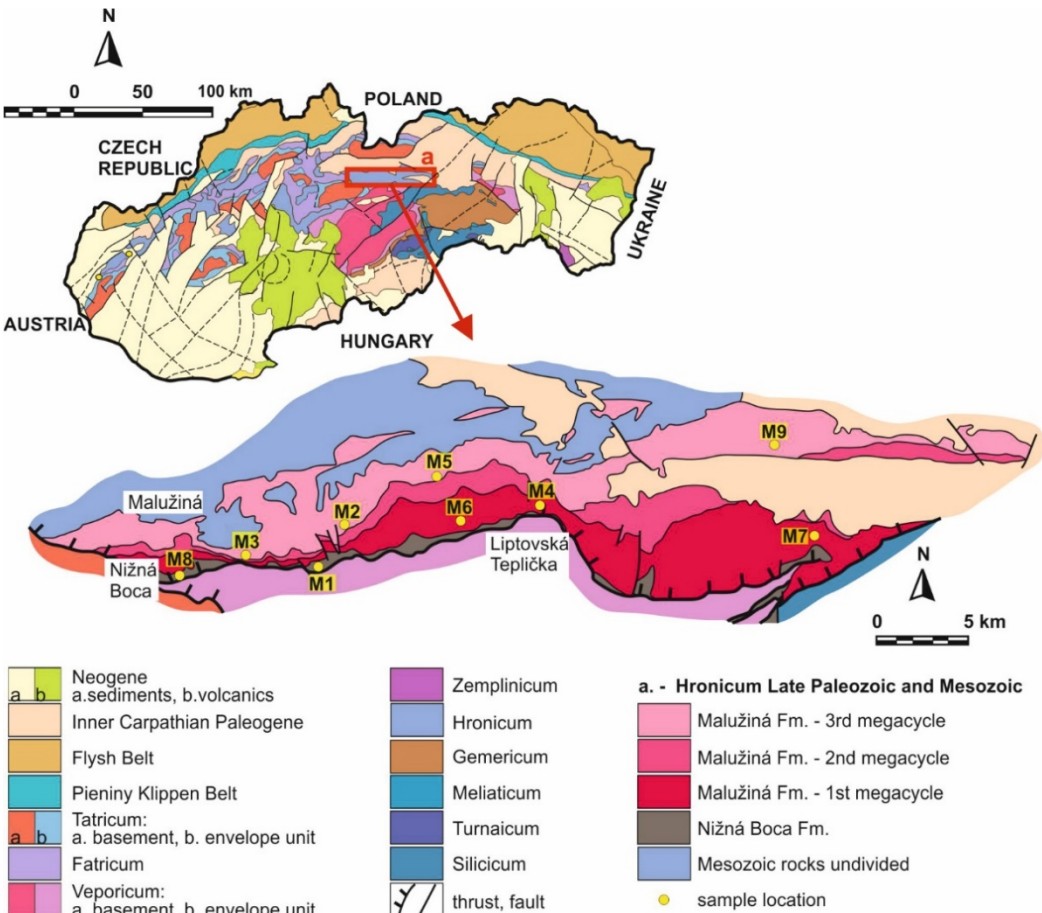

**Figure 1.** Simplified geological sketch map of the Western Carpathians [15] and schematic geological map of the Hronicum Pennsylvanian–Permian sequence on the northern slopes of the Nízke Tatry Mts. showing sample locations [15].

The allochthonous position of Hronicum superficial nappes relative to basement implies an ultra-Veporic origin. Hronicum probably comes from the area that originally existed between the Veporicum and Gemericum units [16–18]. The southernmost tectonic occurrences of Hronic debris were recognized along the line Lubeník–Margecany (tectonic contact Veporicum/Gemericum). Generally, the Hronicum nappes rest directly on the Fatricum unit, except for Veporicum zone where Hronicum directly overthrust the crystalline complex of Veporicum [19].

The original basement of the Hronicum is currently undetermined. The character of the Pennsylvania–Permian terrestrial siliciclastic sediments suggests that they originated from a sedimentary basin located on the continental crust. This opinion is also supported by findings of some mylonitized granitoid tectonic slices below the basal part of the lower partial Hronicum nappe [20,21]. The age of the source area (310–340 Ma) was determined based on the cooling of detrital mica in sandstones (40Ar/39Ar [6]) and suggests that the beginning of the Hronicum sedimentary basin foundation is likely younger than 310 Ma (Moscovian). Analysis of detrital zircon ages indicates the possible derivation of the Hronicum sediments from the Variscan Western Carpathian crystalline basement, mainly the Late Devonian/Early Mississippian magmatic arc (345–370 Ma; [8]). Furthermore, the whole zircon age spectra support close relations between Horonicum and the Armorican terranes and derivation from the Cadomian Belt, which were associated with the West African Craton during the Neoproterozoic and Cambrian time [8], are supported.

The most complete Pennsylvanian–Permian sediments of the Hronic unit have been preserved in the Nízke Tatry Mts (Figure 1). The basal part of the Hronic nappe system is represented by the uppermost Carboniferous–Permian volcano-sedimentary sequence

called the Ipoltica Group (Figure 1). It comprises the uppermost Carboniferous lowermost Permian Nižná Boca Fm. and the Permian Malužiná Fm. [3,22].

Nižná Boca Formation is the lower lithostratigraphic unit and its age was determined based on the macroflora from the upper part of Nižná Boca Fm as the uppermost Pennsylvanian, Kasimovian–Gzhelian (ICS Stratigraphic Chart 2017). According to the regional stratigraphic scale of Central Europe this correlates with Stephanian [23]. This fact was documented by Sitár and Vozár [24] based on well-preserved macroflora and later by Planderová [25] on the basis of palynology. Nižná Boca Fm. represents a cyclic grey-to-black clastic sequence with a distinct tendency of coarsening upwards. It consists of a multiple vertical alteration of regressive lacustrine-deltaic succession. Their unification in upper part indicates mutual prograding from lacustrine delta to fluvial environment. This sedimentary association was interrupted by synsedimentary subaerial volcanism as manifested by abundant redeposited volcanogenic detrital material mixed with non-volcanic detritus. Thin layers of dacite tuffs are present but are less frequent and similarly exceptional small lava flows can be observed [26]. Sporadic doleritic sills and dykes, occurring in the upper part of the formation, are regarded as co-magmatic with the main Permian volcanism of the younger Malužiná Fm.

The Malužiná formation represents a volcanosedimentary complex, consisting of polycyclic sequences from small to large sedimentary cycles. Deposits can be classified as two main alluvial systems (alluvium of braided rivers and meandering rivers). This sequences likely represent red beds type of sediments, the formation of which consists of brightly colored, mainly red clay and sandy shales, arkose sandstones, siltstones, conglomerates and locally also evaporites, which are assembled into upwardly refining cycles. The characteristic feature of the Malužiná Fm. is the extensive andesite–basalt continental tholeitic volcanism, which was generated during two main eruption phases [9–11]. They belong to the 1st and 3rd megacycle. The 2nd megacycle consists mainly of fluvial-alluvial clastic rocks with fining-upward trend. Small portions of effusive and volcaniclastic rocks occur at the base of the 2nd megacycle.

The Autunian–Saxonian microflora assemblages, described by Planderová [27] and Planderová and Vozárová [28], from the Malužiná Fm. 1st and 2nd megacycles correspond to the Cisuralian–Guadalupian, according to the Global Stratigraphic Scale (ICS, 2017) and the stratigraphic correlations by Izart et al. [29]. This assumption is supported by the Pb/U dating from uranium-bearing layers that have been found within the 2nd mega-cycle ($263 \pm 11$ Ma, [30]). The important magnetostratigraphic data, confirming the position of the Illawarra Reversal Magnetic Horizon, were obtained from the upper part of the 2nd megacycle. Menning [31,32] proved that this reversal magnetic event occurred at the 265 Ma transition. On the basis of magnetostratigraphic results, as well as the Thuringian microflora described by Planderová [27], the 3rd megacycle is considered to be of the Lopingian age.

Commonly, the grade of regional metamorphism did not exceed the very low-grade boundary (according to pumpellyite+prehnite+quartz assemblage–Vrána and Vozár [33]; illite crystallinity indexes from pelites—Plašienka et al. [34] and Šucha and Eberl [35]).

## 3. Analytical Methods

Silicates were studied using electron microprobe JEOL JXA 8530FE at the Earth Sciences Institute in Banská Bystrica under the following conditions: accelerating voltage 15 kV, probe current 20 nA, beam diameter 3–8 μm, ZAF correction, counting time 10 s on peak, 5 s on background. The used standards, X-ray lines, and detection limit (in ppm) were: Ca(Kα, 25)—diopside, K (Kα, 44)—orthoclase, P (Kα, 41)—apatite, F (Kα, 167)—fluorite, Na (Kα, 43)—albite, Mg (Kα, 41)—diopside, Al (Kα, 42)—albite, Si (Kα, 63)—quartz, Ba (Lα, 72)—barite, Fe (Kα, 52)—hematite, Cr (Kα, 113)—Cr2O3, Mn (Kα, 59)—rhodonite, Ti (Kα, 130)—rutile, Cl (Kα, 12)—tugtupite, Sr (Lα, 71)—celestite.

The chemical composition of the rocks was determined at the ACME Analytical Laboratories (Vancouver, BC, Canada). Total abundances of major element oxides were

determined by inductively coupled plasma–emission spectrometry (ICP–ES) following lithium metaborate–tetraborate fusion and dilute nitric acid treatment. Loss on ignition (LOI) was calculated from the difference in weight after ignition to 1000 °C. For the total carbon (TOT/C) and sulfur analysis (TOT/S) by LECO analysis, the samples were heated in an induction furnace to >1650 °C, which caused volatilization of all C and S bearing phases. The vapors were carried through an infrared spectrometric cell wherein the concentrations of C and S were determined by the absorption of specific wavelengths in the infrared spectra (ORG/C = TOT/C minus graphite C and carbonate). The concentrations of trace elements and rare earth elements were determined by ICP mass spectrometry (ICP-MS). Further details are accessible on the web page of the ACME Analytical Laboratories (http://acmelab.com/, accessed on 9 November 2019).

The samples were analyzed for Pb and Sr isotopes using a VG Sector 54 IT thermal ionization mass spectrometer (TIMS) at the Department of Geosciences and Natural Resource Management, University of Copenhagen, Denmark. The samples were dissolved in concentrated $HNO_3$, HCl and HF within Savillex™ beakers on a hotplate at 130 °C for 3 days [36,37]. Chromatographic separation columns charged with 12 mL AG50W-X 8 (100–200 mesh) cation resin were used for Sr separation. Strontium fractions were purified using a standardized 3 M $HNO_3$–$H_2O$ elution recipe on self-made disposable mini-extraction columns including mesh 50–100 SrSpec™ (Eichrom Inc., Lisle, IL, USA) resin. Separation of Pb was performed using conventional glass and miniature glass stem anion exchange columns containing 1 mL and 200 μL of 100–200 mesh Bio-Rad AG 1 × 8 resin, respectively. Lead isotopes were measured in a static multi-collection mode and fractionation was controlled by repeated analysis of the NBS 981 standard (using values of [38]. Total procedural blanks were below 200 pg Pb, which do not influence lead isotope ratios beyond the significant third digit. Hundred ng loads of the NBS 987 Sr standard gave $^{87}Sr/^{86}Sr = 0.710238 \pm 0.000018$ ($n = 5$, $2\sigma$). The $^{87}Sr/^{86}Sr$ values of the samples were corrected for the offset relative to the certified NIST SRM 987 value of 0.710245 [39].

Apatite crystals were separated using standard techniques. Apatite U-Pb data were acquired using a Photon Machines Analyte Exite 193 nm ArF Excimer laser-ablation system coupled to a Thermo Scientific iCAP Qc at the Department of Geology Trinity College Dublin. Twenty-eight isotopes ($^{31}P$, $^{35}Cl$, $^{43}Ca$, $^{55}Mn$, $^{86}Sr$, $^{89}Y$, $^{139}La$, $^{140}Ce$, $^{141}Pr$, $^{146}Nd$, $^{147}Sm$, $^{153}Eu$, $^{157}Gd$, $^{159}Tb$, $^{163}Dy$, $^{165}Ho$, $^{166}Er$, $^{169}Tm$, $^{172}Yb$, $^{175}Lu$, $^{200}Hg$, $^{204}Pb$, $^{206}Pb$, $^{207}Pb$, $^{208}Pb$, $^{232}Th$, $^{238}U$, and mass ($^{248/232}Th\ ^{16}O$) were acquired using a 50 μm laser spot, a 4 Hz laser repetition rate, and a fluence of 3.31 J/cm2. A ca. 1 cm sized crystal of Madagascar apatite, which has yielded a weighted average ID-TIMS concordia age of 473.5 ± 0.7 Ma [40,41], was used as the primary apatite reference material in this study. McClure Mountain syenite apatite (the rock from which the $^{40}Ar/^{39}Ar$ hornblende standard MMhb is derived) was used as a secondary standard. McClure Mountain syenite has moderate but reasonably consistent U and Th contents (~23 ppm and 71 ppm—[42]) and its thermal history, crystallization age (weighted mean $^{207}Pb/^{235}U$ age of 523.51 ± 2.09 Ma), and initial Pb isotopic composition ($^{206}Pb/^{204}Pb = 17.54 \pm 0.24$; $^{207}Pb/^{204}Pb = 15.47 \pm 0.04$) are known from high-precision TIMS analyses [43]. Durango apatite was also analyzed in this study as a secondary standard. Durango apatite is a distinctive yellow-green fluorapatite widely used as a mineral standard in apatite fission-track and (U-Th)/He dating and apatite electron micro-probe analyses. It is found as large crystals within an open pit iron mine at Cerro de Mercado, Durango, Mexico. The apatite formed between the eruptions of two major ignimbrites, which have yielded a sanidine-anorthoclase $^{40}Ar$—$^{39}Ar$ age of 31.44 ± 0.18 Ma [44]. NIST 612 standard glass was used as the apatite trace element concentration reference material. The raw isotope data were reduced using the "VizsualAge" data reduction scheme of Petrus and Kamber [45] within the freeware IOLITE package of Paton et al. [46]. User-defined time intervals are established for the baseline correction procedure to calculate session-wide baseline-corrected values for each isotope. The time-resolved fractionation response of individual standard analyses is then characterized using a user-specified down-hole correction model (such as an exponential curve, a linear fit, or a

smoothed cubic spline). The data reduction scheme then fits this appropriate sessionwide "model" U-Th-Pb fractionation curve to the time-resolved standard data and the unknowns. Sample-standard bracketing is applied after the correction of down-hole fractionation to account for long-term drift in isotopic or elemental ratios by normalizing all ratios to those of the U-Pb reference standards. Common Pb in the apatite standards was corrected using the [207]Pb-based correction method using a modified version of the VizualAge DRS that accounts for the presence of variable common Pb in the primary standard materials [47]. Over the course of two months of analyses, McClure Mountain apatite ([207]Pb/[235]U TIMS age of 523.51 ± 1.47 Ma—[43]) yielded a U-Pb Tera-Wasserburg concordia lower intercept age of 524.5 ± 3.7 Ma with an MSWD = 0.72. The lower intercept was anchored using a [207]Pb/[206]Pb value of 0.88198 derived from apatite ID-TIMS total U-Pb isochron [43].

The structural formulas of the minerals were calculated using standard normalization procedure, i.e., clinopyroxenes on 6 oxygens, feldspars on 8 oxygens, spinels on 4 oxygens, and biotite on equivalent of 22 oxygens. $Fe^{2+}$ and $Fe^{3+}$ were calculated based on charge balance. The structural formulas of amphiboles were recalculated on 24 anions using the Amphibole classification Excel spreadsheet v1.9.3 [48].

## 4. Petrography and Mineralogy

The rocks studied are strongly altered and the primary mineral composition is preserved only in some localities. Most of the aphanitic and fine-grained volcanites are strongly chloritized and, in places, carbonated. The coarser-grained varieties of rocks are less altered, in places. Clinopyroxene, plagioclase, and amphibole are locally preserved from the primary minerals. In basaltic andesites and basalts of the 1st and 2nd eruption phase, the predominant textures are intersertal, pilotaxitic, and hyalopilitic, which are characteristic of fine-grained varieties. In relatively coarse-grained varieties, typical for the first eruption phase, the common textures are trachytic, fine-grained porphyritic, and hyaloophitic. In glassy varieties, volcanic glass is devitrified and rich in Fe-Ti oxides. Amygdaloid and vesicular structures have been commonly identified. In the coarser-grained diorite dykes and sills, the ophitic texture is well preserved.

In fine-grained rocks, pyroxene forms hypidiomorphic phenocrysts, but they are completely altered to a mixture of chlorite and carbonate. Its identification is possible only on the basis of crystal shape within pseudomorphoses. The matrix in basalts and basalt andesites is composed of crystallites of plagioclase, devitrified glass, abundant Fe-Ti oxides, and chlorite pseudomorphoses after small crystals of clinopyroxenes. Clinopyroxene in medium- and coarse-grained rocks forms skeletal individuals and intimately overgrows with plagioclase (Figure 2a,c,e,f) within the ophitic texture. The clinopyroxenes are relatively homogenous in chemical composition, with only rarely observed darker zones in larger grains (Figure 2e) rich in Mg, and/or depleted of Fe. However, this is not typical oscillatory or sector zoning. The chemical composition of clinopyroxenes is characterized by relatively low contents of $TiO_2$ (0.97–1.56 wt.%) and alkalis (0.26–0.34 wt.%) (Table 2). The Cpx from M-8 sample are slightly different; they have higher $TiO_2$ a MgO, and or lower FeO and $Al_2O_3$ contents. It is probably a reflection of the parent rocks composition (Cr-spinels were also identified in M-8 sample). Based on the classification of pyroxenes (Figure 3), these Cpx correspond to augite. Clinopyroxenes are often lined with younger magmatic amphiboles (Figure 2a,b). Similarly, allotriomorphic grains of ilmenites are observed at the edge of clinopyroxenes. Alternatively, these may represent possible exsolutions of $TiO_2$ from the peripheral parts of the clinopyroxenes.

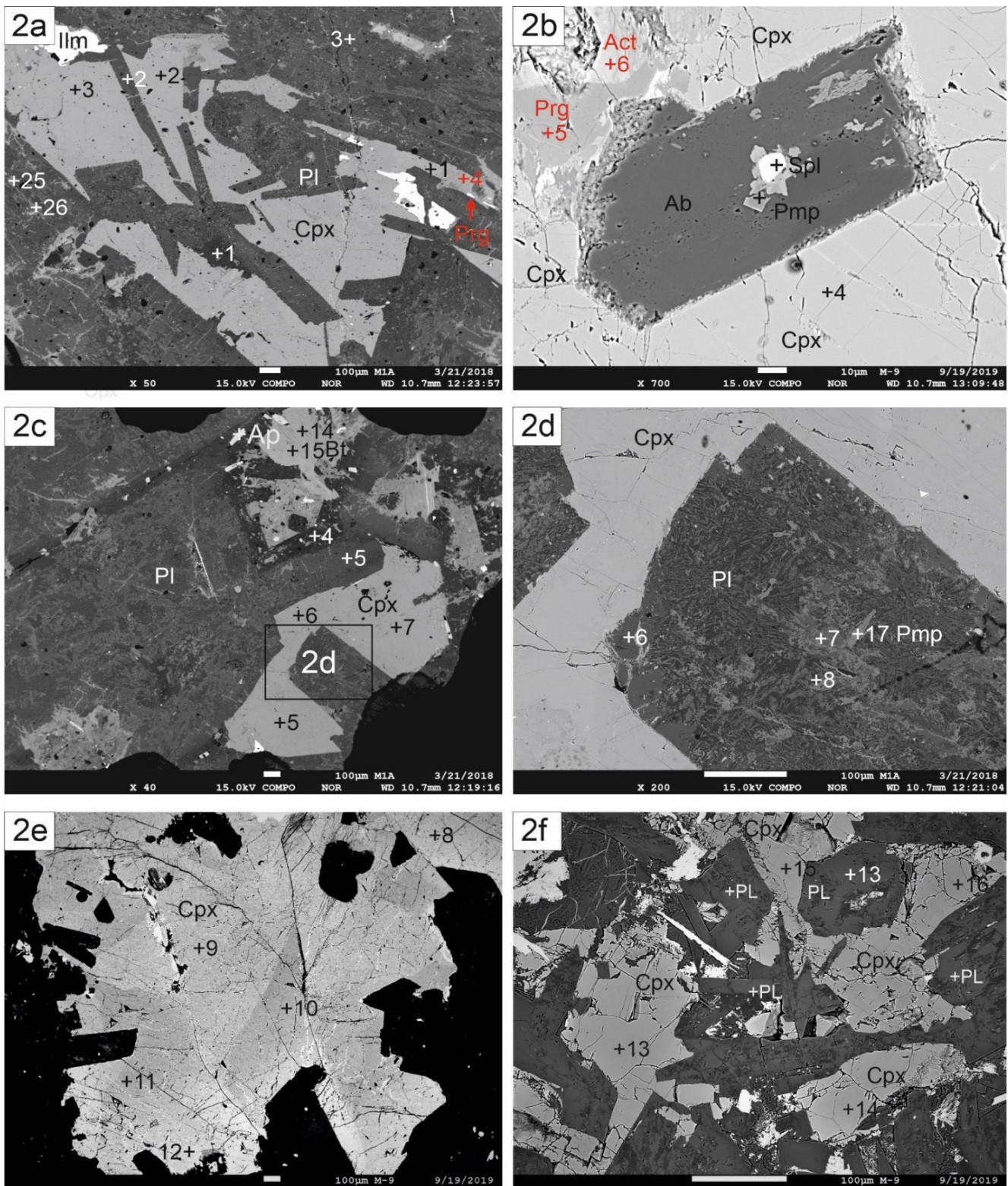

**Figure 2.** Back scattered electron (BSE) images of basalts from Malužiná; (**a**)—overgrowth Pl and Cpx, (**b**)—porphyric Pl around Cpx; (**c**)—altered Pl, (**d**)—detail of alterered Pl with Pmp, (**e**)—zonal porphyric Cpx, (**f**)—texture of basalt with Pl and Cpx; Cpx = clinopyroxene, Prg = pargasite, Act = actinolite, Pl = plagioclase, Bt = biotite, Pmp = pumpellyite, Ilm = ilmenite; the numbers in figures correspond to those in Tables 2–5.

**Table 2.** Selected analyses of clinopyroxenes.

| Sample | M-1 | | M-8 | | | M-1 | | | | M-8 | | | | | M-9 | | | |
|---|---|---|---|---|---|---|---|---|---|---|---|---|---|---|---|---|---|---|
| Figures | 2a | | 2b | | | 2c | | | | 2e | | | | | 2f | | | |
| Anal. N. | 1 | 2 | 3 | 4 | 5 | 6 | 7 | 8 | 9 | 10 | 11 | 12 | 13 | 14 | 15 | 16 | 17 | 18 |
| $SiO_2$ | 51.74 | 50.34 | 50.46 | 52.08 | 50.23 | 50.15 | 50.84 | 50.23 | 50.50 | 51.84 | 51.82 | 51.15 | 50.97 | 50.78 | 50.43 | 50.73 | 50.92 | 50.83 |
| $TiO_2$ | 1.05 | 1.53 | 1.56 | 1.97 | 1.45 | 1.58 | 0.97 | 3.34 | 3.41 | 2.33 | 2.15 | 1.82 | 1.00 | 1.27 | 1.35 | 1.19 | 1.31 | 0.80 |
| $Al_2O_3$ | 1.65 | 2.60 | 2.63 | 0.71 | 2.69 | 2.90 | 2.13 | 1.09 | 0.80 | 0.68 | 0.78 | 0.79 | 2.29 | 2.13 | 2.39 | 2.13 | 2.34 | 2.40 |
| $Cr_2O_3$ | 0.03 | 0.00 | 0.08 | 0.50 | 0.00 | 0.16 | 0.05 | 0.83 | 0.93 | 0.37 | 0.09 | 0.00 | 0.23 | 0.08 | 0.14 | 0.04 | 0.17 | 0.52 |
| $FeO^+$ | 10.87 | 10.21 | 9.91 | 5.90 | 9.68 | 9.31 | 11.23 | 6.03 | 6.04 | 4.57 | 6.61 | 8.95 | 10.65 | 10.43 | 9.63 | 10.57 | 10.03 | 9.28 |
| MnO | 0.24 | 0.30 | 0.17 | 0.13 | 0.30 | 0.23 | 0.24 | 0.11 | 0.20 | 0.09 | 0.10 | 0.26 | 0.22 | 0.23 | 0.18 | 0.16 | 0.09 | 0.23 |
| MgO | 13.37 | 14.24 | 14.10 | 17.60 | 13.98 | 13.78 | 13.55 | 16.28 | 16.62 | 17.06 | 16.83 | 15.89 | 16.60 | 15.41 | 15.64 | 14.74 | 15.83 | 17.04 |
| CaO | 19.75 | 19.53 | 19.06 | 20.80 | 19.28 | 19.77 | 19.40 | 21.35 | 21.13 | 21.24 | 21.30 | 20.28 | 17.35 | 19.11 | 19.18 | 19.51 | 18.86 | 17.35 |
| $Na_2O$ | 0.26 | 0.27 | 0.32 | 0.19 | 0.33 | 0.34 | 0.32 | 0.25 | 0.22 | 0.18 | 0.18 | 0.25 | 0.25 | 0.22 | 0.27 | 0.23 | 0.21 | 0.22 |
| $K_2O$ | 0.00 | 0.00 | 0.00 | 0.01 | 0.00 | 0.00 | 0.00 | 0.00 | 0.00 | 0.00 | 0.01 | 0.00 | 0.01 | 0.00 | 0.01 | 0.00 | 0.00 | 0.01 |
| Total | 98.96 | 99.02 | 98.28 | 99.88 | 97.93 | 98.22 | 98.73 | 99.51 | 99.84 | 98.36 | 99.88 | 99.38 | 99.57 | 99.63 | 99.22 | 99.29 | 99.75 | 98.69 |
| Formula based on 6 oxygens | | | | | | | | | | | | | | | | | | |
| Si | 1.964 | 1.899 | 1.917 | 1.916 | 1.914 | 1.906 | 1.931 | 1.870 | 1.874 | 1.936 | 1.914 | 1.911 | 1.896 | 1.897 | 1.886 | 1.906 | 1.895 | 1.899 |
| Ti | 0.030 | 0.043 | 0.045 | 0.054 | 0.042 | 0.045 | 0.028 | 0.094 | 0.095 | 0.065 | 0.060 | 0.051 | 0.028 | 0.036 | 0.038 | 0.034 | 0.037 | 0.023 |
| Al | 0.074 | 0.115 | 0.118 | 0.031 | 0.121 | 0.130 | 0.095 | 0.048 | 0.035 | 0.030 | 0.034 | 0.035 | 0.101 | 0.094 | 0.105 | 0.094 | 0.102 | 0.106 |
| Cr | 0.001 | 0.000 | 0.002 | 0.028 | 0.000 | 0.005 | 0.001 | 0.019 | 0.016 | 0.000 | 0.029 | 0.059 | 0.063 | 0.055 | 0.063 | 0.042 | 0.044 | 0.052 |
| $Fe^{3+}$ | 0.000 | 0.019 | 0.000 | 0.015 | 0.000 | 0.000 | 0.010 | 0.024 | 0.027 | 0.011 | 0.003 | 0.000 | 0.007 | 0.002 | 0.004 | 0.001 | 0.005 | 0.015 |
| $Fe^{2+}$ | 0.345 | 0.303 | 0.315 | 0.153 | 0.308 | 0.296 | 0.347 | 0.169 | 0.171 | 0.143 | 0.175 | 0.221 | 0.268 | 0.271 | 0.238 | 0.290 | 0.268 | 0.238 |
| Mn | 0.008 | 0.010 | 0.006 | 0.004 | 0.010 | 0.008 | 0.008 | 0.003 | 0.006 | 0.003 | 0.003 | 0.008 | 0.007 | 0.007 | 0.006 | 0.005 | 0.003 | 0.007 |
| Mg | 0.757 | 0.801 | 0.798 | 0.965 | 0.794 | 0.781 | 0.767 | 0.903 | 0.919 | 0.950 | 0.927 | 0.885 | 0.921 | 0.858 | 0.872 | 0.826 | 0.879 | 0.949 |
| Ca | 0.803 | 0.789 | 0.776 | 0.820 | 0.787 | 0.805 | 0.790 | 0.851 | 0.840 | 0.850 | 0.843 | 0.812 | 0.691 | 0.765 | 0.768 | 0.786 | 0.752 | 0.694 |
| Na | 0.019 | 0.020 | 0.023 | 0.014 | 0.024 | 0.025 | 0.024 | 0.018 | 0.016 | 0.013 | 0.013 | 0.018 | 0.018 | 0.016 | 0.019 | 0.017 | 0.015 | 0.016 |
| K | 0.000 | 0.000 | 0.000 | 0.000 | 0.000 | 0.000 | 0.000 | 0.000 | 0.000 | 0.000 | 0.000 | 0.000 | 0.000 | 0.000 | 0.000 | 0.000 | 0.000 | 0.000 |
| Wo% | 42.16 | 41.27 | 41.08 | 42.30 | 41.65 | 42.77 | 41.27 | 44.27 | 43.51 | 43.76 | 43.34 | 42.33 | 36.77 | 40.38 | 40.91 | 41.32 | 39.62 | 36.91 |
| En% | 39.72 | 41.88 | 42.26 | 49.80 | 42.03 | 41.50 | 40.09 | 46.97 | 47.61 | 48.89 | 47.66 | 46.15 | 48.96 | 45.32 | 46.42 | 43.42 | 46.27 | 50.44 |
| Fs% | 18.12 | 16.85 | 16.66 | 7.90 | 16.32 | 15.73 | 18.64 | 8.76 | 8.88 | 7.35 | 9.00 | 11.51 | 14.27 | 14.30 | 12.66 | 15.26 | 14.11 | 12.65 |

$FeO^+$ = total Fe as FeO.



**Table 3.** Selected analyses of amphiboles.

| Sample | M-1 | | | | | M-8 | | | | | | |
|---|---|---|---|---|---|---|---|---|---|---|---|---|
| Anal. N. | 1 | 2 | 3 | 4 | 5 | 6 | 7 | 8 | 9 | 10 | 11 | 12 |
| Figures | | | | 2a | | 2b | | | | | | |
| $SiO_2$ | 46.77 | 47.14 | 46.23 | 44.03 | 45.73 | 54.07 | 52.98 | 53.19 | 52.62 | 52.75 | 52.05 | 51.92 |
| $TiO_2$ | 0.20 | 0.40 | 0.52 | 2.79 | 0.42 | 0.13 | 0.07 | 0.01 | 0.00 | 0.00 | 0.01 | 0.04 |
| $Al_2O_3$ | 8.55 | 8.43 | 8.89 | 8.86 | 9.85 | 0.61 | 1.58 | 1.74 | 1.44 | 1.52 | 1.68 | 2.05 |
| $Cr_2O_3$ | 0.05 | 0.00 | 0.06 | 0.00 | 0.22 | 0.05 | 0.08 | 0.01 | 0.03 | 0.03 | 0.08 | 0.03 |
| $FeO^+$ | 9.10 | 9.23 | 9.61 | 13.41 | 8.38 | 14.57 | 16.32 | 13.49 | 16.60 | 16.63 | 17.49 | 17.54 |
| MnO | 0.09 | 0.15 | 0.14 | 0.18 | 0.05 | 0.35 | 0.45 | 0.49 | 0.49 | 0.39 | 0.60 | 0.55 |
| MgO | 16.91 | 16.47 | 16.51 | 13.27 | 18.33 | 14.66 | 13.84 | 15.82 | 13.33 | 13.34 | 12.35 | 12.14 |
| CaO | 11.16 | 11.72 | 11.40 | 10.62 | 11.75 | 12.44 | 11.80 | 11.59 | 12.00 | 12.25 | 12.13 | 11.97 |
| $Na_2O$ | 2.06 | 1.95 | 2.13 | 2.28 | 2.94 | 0.11 | 0.22 | 0.36 | 0.16 | 0.15 | 0.23 | 0.32 |
| $K_2O$ | 0.42 | 0.34 | 0.45 | 0.53 | 0.17 | 0.04 | 0.04 | 0.02 | 0.04 | 0.04 | 0.06 | 0.05 |
| Total | 95.29 | 95.83 | 95.93 | 95.96 | 97.83 | 97.02 | 97.37 | 96.71 | 96.70 | 97.12 | 96.69 | 96.59 |
| Formula based on 24 (O, OH, F, Cl,) | | | | | | | | | | | | |
| Si | 6.824 | 6.867 | 6.738 | 6.588 | 6.496 | 7.898 | 7.756 | 7.724 | 7.776 | 7.759 | 7.740 | 7.731 |
| AlIV | 1.176 | 1.133 | 1.262 | 1.412 | 1.504 | 0.102 | 0.244 | 0.276 | 0.224 | 0.241 | 0.260 | 0.269 |
| T | 8.000 | 8.000 | 8.000 | 8.000 | 8.000 | 8.000 | 8.000 | 8.000 | 8.000 | 8.000 | 8.000 | 8.000 |
| Ti | 0.022 | 0.043 | 0.057 | 0.314 | 0.045 | 0.014 | 0.007 | 0.001 | 0.000 | 0.000 | 0.001 | 0.005 |
| AlVI | 0.294 | 0.315 | 0.265 | 0.150 | 0.145 | 0.003 | 0.028 | 0.021 | 0.026 | 0.023 | 0.035 | 0.090 |
| Cr | 0.006 | 0.000 | 0.006 | 0.000 | 0.025 | 0.006 | 0.010 | 0.002 | 0.003 | 0.004 | 0.010 | 0.003 |
| $Fe^{3+}$ | 0.445 | 0.300 | 0.425 | 0.217 | 0.631 | 0.056 | 0.184 | 0.247 | 0.188 | 0.206 | 0.202 | 0.156 |
| $Mn^{2+}$ | 0.000 | 0.000 | 0.000 | 0.000 | 0.000 | 0.006 | 0.000 | 0.000 | 0.000 | 0.002 | 0.042 | 0.024 |
| $Fe^{2+}$ | 0.556 | 0.764 | 0.661 | 1.360 | 0.272 | 1.723 | 1.752 | 1.305 | 1.847 | 1.839 | 1.973 | 2.027 |
| Mg | 3.677 | 3.578 | 3.586 | 2.959 | 3.882 | 3.192 | 3.020 | 3.424 | 2.936 | 2.926 | 2.737 | 2.694 |
| C | 5.000 | 5.000 | 5.000 | 5.000 | 5.000 | 5.000 | 5.001 | 5.000 | 5.000 | 5.000 | 5.000 | 4.999 |
| $Mn^{2+}$ | 0.011 | 0.019 | 0.017 | 0.023 | 0.006 | 0.037 | 0.056 | 0.061 | 0.061 | 0.047 | 0.034 | 0.045 |
| $Fe^{2+}$ | 0.109 | 0.061 | 0.086 | 0.101 | 0.093 | 0.000 | 0.063 | 0.086 | 0.017 | 0.000 | 0.000 | 0.000 |
| Ca | 1.745 | 1.829 | 1.781 | 1.702 | 1.788 | 1.947 | 1.851 | 1.803 | 1.900 | 1.931 | 1.933 | 1.909 |
| Na | 0.136 | 0.091 | 0.117 | 0.173 | 0.113 | 0.016 | 0.031 | 0.051 | 0.023 | 0.022 | 0.033 | 0.046 |
| B | 2.001 | 2.000 | 2.001 | 1.999 | 2.000 | 2.000 | 2.001 | 2.001 | 2.001 | 2.000 | 2.000 | 2.000 |
| Na | 0.446 | 0.459 | 0.485 | 0.488 | 0.696 | 0.016 | 0.031 | 0.051 | 0.023 | 0.022 | 0.034 | 0.046 |
| K | 0.077 | 0.063 | 0.084 | 0.101 | 0.031 | 0.008 | 0.008 | 0.004 | 0.007 | 0.008 | 0.012 | 0.009 |
| A | 0.523 | 0.522 | 0.569 | 0.589 | 0.727 | 0.024 | 0.039 | 0.055 | 0.030 | 0.030 | 0.046 | 0.055 |

$FeO^+$ = total Fe as FeO.

**Table 4.** Selected analyses of biotites, prehnites, and pumpellyites.

| Sample | M1 | | M-1 | | | | | M-8 | |
|---|---|---|---|---|---|---|---|---|---|
| Mineral | Biotite | | Pumpellyite | | | | | Prehnite | |
| Anal. N. | 14 | 15 | 17 | 25 | 26 | 2 | 22 | 24 | 27 |
| $SiO_2$ | 37.69 | 37.96 | 36.84 | 37.26 | 37.33 | 37.95 | 37.29 | 43.56 | 43.45 |
| $TiO_2$ | 4.58 | 4.48 | 0.02 | 0.16 | 0.06 | 0.13 | 0.02 | 0.10 | 0.18 |
| $Al_2O_3$ | 12.18 | 12.15 | 25.90 | 25.37 | 25.12 | 29.85 | 26.89 | 24.52 | 24.51 |
| $Cr_2O_3$ | 0.00 | 0.00 | | | | | | | |
| $FeO^+$ | 16.55 | 17.19 | 3.07 | 3.33 | 3.08 | 0.71 | 2.47 | 0.27 | 0.15 |
| MnO | 0.04 | 0.00 | 0.14 | 0.08 | 0.00 | 0.04 | 0.15 | 0.00 | 0.00 |
| MgO | 13.40 | 13.11 | 1.71 | 2.29 | 2.28 | 0.09 | 1.98 | 0.00 | 0.03 |
| CaO | 0.00 | 0.03 | 22.10 | 22.30 | 22.36 | 23.45 | 23.31 | 27.36 | 27.27 |
| BaO | 0.17 | 1.43 | 0.01 | 0.00 | 0.00 | 0.00 | 0.00 | 0.00 | 0.00 |
| SrO | 0.00 | 0.00 | 0.02 | 0.01 | 0.00 | 0.00 | 0.00 | 0.00 | 0.00 |
| $Na_2O$ | 0.14 | 0.17 | 0.13 | 0.04 | 0.01 | 0.03 | 0.03 | 0.06 | 0.02 |
| $K_2O$ | 9.05 | 9.06 | 0.02 | 0.00 | 0.00 | 0.00 | 0.04 | 0.02 | 0.02 |
| Cl | 0.09 | 0.11 | | | | | | | |
| F | 0.19 | 0.00 | | | | | | | |
| Total | 94.07 | 95.67 | 89.94 | 90.83 | 90.23 | 92.26 | 92.18 | 95.88 | 95.63 |
| | Formula based on 24 (O, OH, Cl, F) | | Formula based on16 cations and 24, 5 oxygens | | | | | Formula based on 22 O * | |
| Si | 5.773 | 5.776 | 6.076 | 6.093 | 6.135 | 6.007 | 5.997 | 5.977 | 5.975 |
| Al IV | 2.199 | 2.179 | 5.035 | 4.891 | 4.866 | 5.568 | 5.098 | 3.965 | 3.972 |
| Al VI | 0.000 | 0.000 | | | | | | | |
| Ti | 0.528 | 0.512 | 0.003 | 0.019 | 0.007 | 0.016 | 0.003 | 0.011 * | 0.019 * |
| Cr | 0.000 | 0.000 | | | | | | | |
| Fe | 2.121 | 2.188 | 0.423 | 0.455 | 0.423 | 0.095 | 0.332 | 0.031 | 0.017 |
| Mn | 0.005 | 0.000 | 0.019 | 0.010 | 0.000 | 0.005 | 0.020 | 0.000 | 0.000 |
| Mg | 3.060 | 2.974 | 0.420 | 0.559 | 0.557 | 0.021 | 0.475 | 0.000 | 0.005 |
| Ca | 0.000 | 0.004 | 3.906 | 3.908 | 3.937 | 3.976 | 4.018 | 4.022 | 4.019 |
| Na | 0.042 | 0.049 | 0.041 | 0.013 | 0.003 | 0.008 | 0.009 | 0.016 | 0.006 |
| K | 1.768 | 1.758 | 0.003 | 0.000 | 0.000 | 0.001 | 0.009 | 0.003 | 0.004 |
| Sr | 0.000 | 0.000 | | | | | | | |
| Ba | 0.010 | 0.085 | | | | | | | |
| OH * | 3.884 | 3.972 | | | | | | | |
| F | 0.092 | 0.000 | | | | | | | |
| Cl | 0.024 | 0.028 | | | | | | | |
| TOTAL | 19.505 | 19.525 | | | | | | | |

$FeO^+$ total Fe as FeO; * total Fe in prehnite is assumed to be $Fe^{3+}$.

**Table 5.** Selected analyses of plagioclases.

| Sample | | | | M1A | | | | | | | | M-9 | | | |
|---|---|---|---|---|---|---|---|---|---|---|---|---|---|---|---|
| Anal. N. | 1 | 2 | 3 | 4 | 5 | 6 | 7 | 8 | 9 | 10 | 11 | 12 | 13 | 14 | 15 |
| | 22 | 23 | 24 | 20 | 21 | 16 | 18 | 19 | 1 | 3 | 4 | 5 | 6 | 7 | 12 |
| Figures | | 2a | | | 2c | | 2d | | | | | | | | |
| $SiO_2$ | 68.92 | 56.18 | 55.06 | 68.67 | 57.82 | 56.34 | 54.01 | 64.48 | 55.14 | 55.27 | 55.78 | 55.27 | 56.26 | 61.22 | 57.38 |
| $TiO_2$ | 0.00 | 0.20 | 0.13 | 0.03 | 0.19 | 0.07 | 0.19 | 0.06 | 0.24 | 0.11 | 0.17 | 0.19 | 0.17 | 0.07 | 0.24 |
| $Al_2O_3$ | 19.21 | 26.84 | 27.75 | 19.32 | 25.81 | 26.66 | 28.35 | 21.62 | 27.75 | 28.12 | 27.73 | 27.86 | 27.33 | 24.18 | 26.45 |
| $FeO^+$ | 0.10 | 0.43 | 0.21 | 0.11 | 0.19 | 0.39 | 0.35 | 0.00 | 0.50 | 0.43 | 0.45 | 0.49 | 0.55 | 0.57 | 0.50 |
| CaO | 0.10 | 9.34 | 10.34 | 0.09 | 8.12 | 9.40 | 10.99 | 2.80 | 10.57 | 11.04 | 10.71 | 10.58 | 10.19 | 6.02 | 9.34 |
| BaO | 0.01 | 0.00 | 0.03 | 0.01 | 0.03 | 0.06 | 0.00 | 0.00 | 0.00 | 0.02 | 0.00 | 0.01 | 0.03 | 0.02 | 0.00 |
| $Na_2O$ | 11.41 | 5.79 | 5.39 | 11.64 | 6.53 | 5.84 | 5.03 | 10.08 | 5.31 | 5.12 | 5.51 | 5.37 | 5.68 | 7.63 | 6.07 |
| $K_2O$ | 0.09 | 0.18 | 0.21 | 0.05 | 0.37 | 0.29 | 0.23 | 0.09 | 0.25 | 0.21 | 0.24 | 0.25 | 0.23 | 0.62 | 0.30 |
| Total | 99.83 | 98.96 | 99.12 | 99.92 | 99.06 | 99.04 | 99.14 | 99.12 | 99.77 | 100.32 | 100.58 | 100.01 | 100.44 | 100.32 | 100.27 |
| Formula based on 5 cations and 8 oxygens | | | | | | | | | | | | | | | |
| Si | 3.022 | 2.556 | 2.506 | 3.004 | 2.619 | 2.562 | 2.461 | 2.862 | 2.496 | 2.490 | 2.500 | 2.494 | 2.523 | 2.725 | 2.574 |
| Ti | 0.000 | 0.007 | 0.005 | 0.001 | 0.006 | 0.002 | 0.006 | 0.002 | 0.008 | 0.004 | 0.006 | 0.006 | 0.006 | 0.002 | 0.008 |
| Al | 0.993 | 1.439 | 1.489 | 0.996 | 1.378 | 1.429 | 1.522 | 1.131 | 1.480 | 1.493 | 1.465 | 1.482 | 1.445 | 1.268 | 1.398 |
| Cr | 0.000 | 0.000 | 0.000 | 0.000 | 0.000 | 0.000 | 0.000 | 0.000 | 0.000 | 0.000 | 0.000 | 0.000 | 0.000 | 0.000 | 0.000 |
| $Fe^{3+}$ | 0.000 | 0.000 | 0.000 | 0.000 | 0.000 | 0.000 | 0.000 | 0.000 | 0.000 | 0.000 | 0.000 | 0.000 | 0.000 | 0.000 | 0.000 |
| $Fe^{2+}$ | 0.004 | 0.016 | 0.008 | 0.004 | 0.007 | 0.015 | 0.013 | 0.000 | 0.019 | 0.016 | 0.017 | 0.019 | 0.021 | 0.021 | 0.019 |
| Ca | 0.004 | 0.455 | 0.504 | 0.004 | 0.394 | 0.458 | 0.537 | 0.133 | 0.512 | 0.533 | 0.514 | 0.511 | 0.490 | 0.287 | 0.449 |
| Ba | 0.000 | 0.000 | 0.000 | 0.000 | 0.001 | 0.001 | 0.000 | 0.000 | 0.000 | 0.000 | 0.000 | 0.000 | 0.000 | 0.000 | 0.000 |
| Na | 0.970 | 0.510 | 0.476 | 0.987 | 0.573 | 0.515 | 0.444 | 0.867 | 0.466 | 0.447 | 0.479 | 0.470 | 0.494 | 0.659 | 0.528 |
| K | 0.005 | 0.010 | 0.012 | 0.003 | 0.021 | 0.017 | 0.013 | 0.005 | 0.015 | 0.012 | 0.014 | 0.014 | 0.013 | 0.035 | 0.017 |
| An | 0.46 | 46.65 | 50.84 | 0.43 | 39.84 | 46.27 | 53.96 | 13.24 | 51.59 | 53.74 | 51.09 | 51.38 | 49.14 | 29.27 | 45.17 |
| Ab | 99.05 | 52.28 | 47.93 | 99.27 | 57.99 | 52.03 | 44.70 | 86.25 | 46.94 | 45.05 | 47.56 | 47.19 | 49.54 | 67.17 | 53.13 |
| Or | 0.49 | 1.07 | 1.22 | 0.29 | 2.17 | 1.69 | 1.34 | 0.51 | 1.47 | 1.21 | 1.35 | 1.43 | 1.32 | 3.57 | 1.70 |

$FeO^+$ total Fe as FeO.

Among other mafic minerals, there occur amphiboles and biotites. Both these minerals are strongly altered. There are two types of amphiboles in the rock (Table 3). It is likely that the older amphibole coexisted with Cpx and has a composition corresponding to pargasite (Figure 4). Pargasite also locally forms rims around Cpx (Figure 2a). The younger amphibole has a composition corresponding to actinolite/tremolite (Figure 4). Compared to the older amphiboles, the younger ones have higher contents of Si, Fe, or lower Mg, Al, and Na.

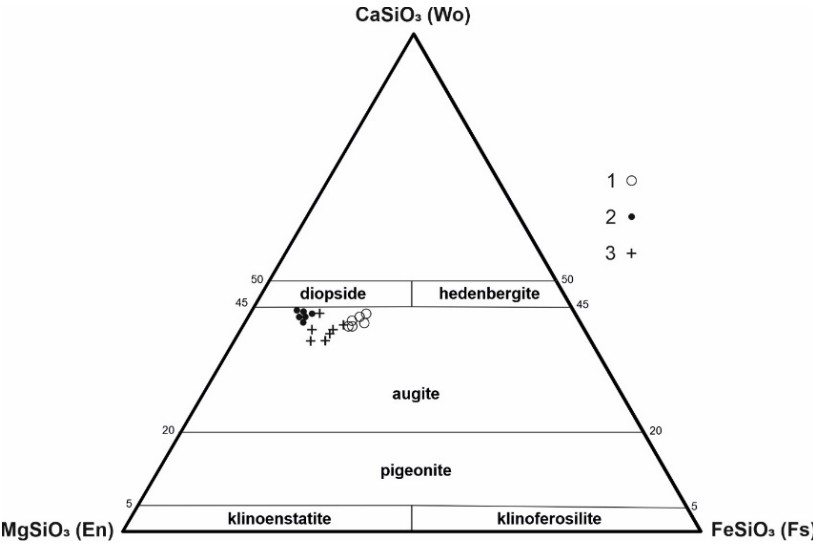

**Figure 3.** Classification diagram of clinopyroxenes [49]. 1—sample M-1; 2—sample M-8; 3—sample M-9.

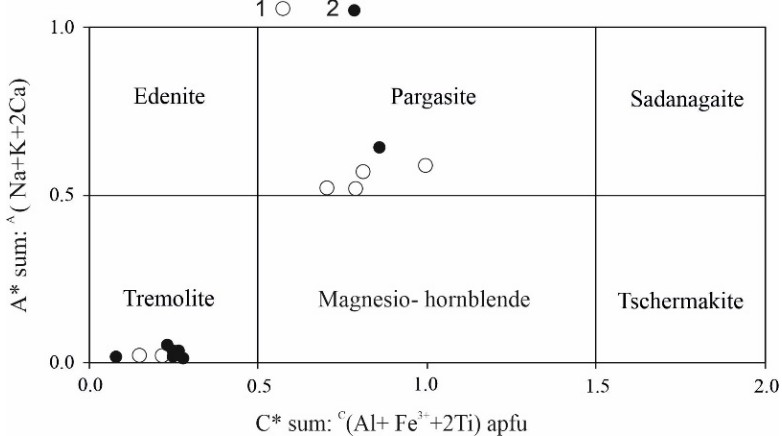

**Figure 4.** The classification diagram of amphiboles, according Hawthorne et al., [50]. 1—sample M-1; 2—sample M-8.

Biotite is strongly altered and pushes amphibole locally at the edges. It is difficult to determine whether it is primary (Table 4), but the high $TiO_2$ suggests it is magmatic biotite. Based on the classification of Rieder et al. [51] (Figure 5A), the biotite has a high content of annite component. The biotite that makes rims around primary amphibole has increased MgO content and thus increased content of phlogopite component (Table 4). Abdel-Rahman [52] used a change in the chemical composition of biotites from different genetic basalt types (Figure 5B). The biotites studied lie in the field of calc-alkaline rocks, which correlates well with the overall chemical composition of the rocks studied.

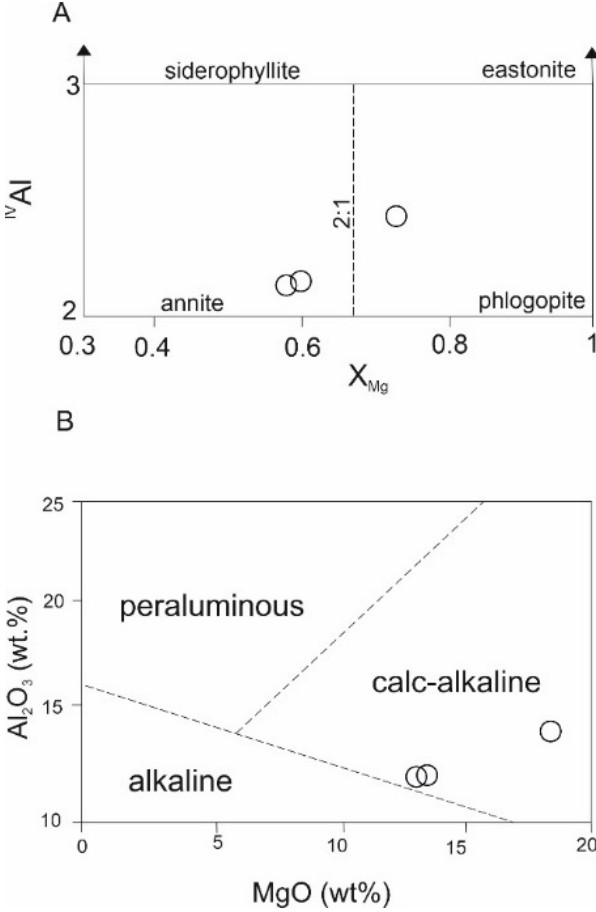

**Figure 5.** (**A**) Classification diagram of studied micas. End-members names according to Rieder et al. [51]; (**B**) Position of studied mica in the classification diagram of Abdel-Rahman [52] for micas from different magmatic series. 1—sample M-1.

Plagioclases in fine-grained strongly altered types, often completely converted to albite, respectively, locally observe tin strips of pumpellyite in the central part of plagioclase (Figure 2a,d). In medium-grained varieties, they are relatively better preserved and relics of primary plagioclases intimately overgrowing with clinopyroxenes can be identified. Overall, in a detailed study, a relatively high inhomogeneity of plagioclases due to post-magmatic and metamorphic processes (Figure 2a,c,d) can be observed. The basicity in the relics of the most preserved primary plagioclases ranges from 40% to 54% of the anorthite component, and thus corresponds to andesine to labradorite (Figure 6, Table 5). In the samples with preserved fresh plagioclase, zoning with anorthite component decreasing toward the plagioclase edge can be observed (Core $An_{49}$, rims $An_{30}$). The composition of altered parts of plagioclase corresponds to albite.

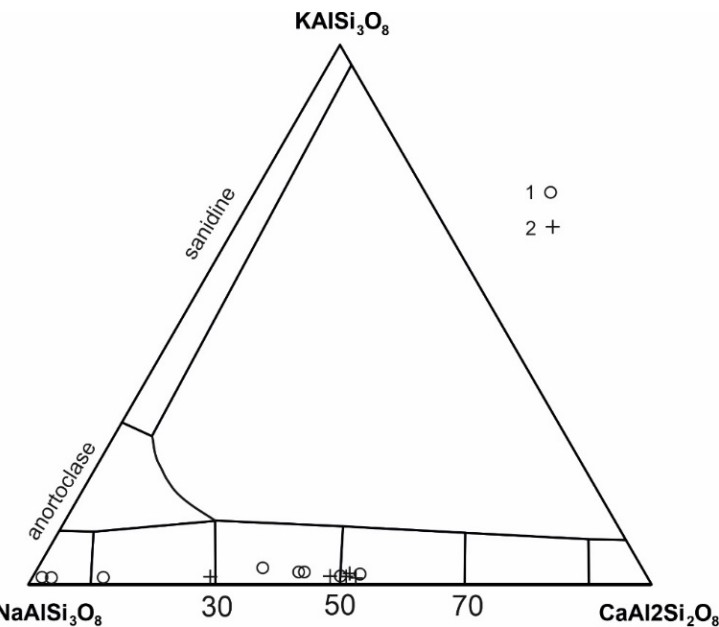

**Figure 6.** Classification diagram of studied plagioclases. 1—sample M-1; 2—sample M-9.

Pumpellyite was detected in the central parts of altered plagioclase and along the fissures of crystals, respectively (Figure 2a,d, Table 4). Pumpellyite and prehnite together with albite form a younger loaded mineral association. At the same time, the presence of pumpellyite and prehnite points to postmagmatic metamorphic processes. Their age and spatial relationships were studied by Vrána and Vozár [33] and Spišiak and Pitoňák [53].

Rare small grains of Cr-spinel have been observed in the gabbro-diorite dyke (sample M-8). Spinel grains probably formed part of the primary mineral association (clinopyroxenes, plagioclases, pargasite). Due to subsequent postmagmatic processes, rocks were altered, and spinels are found in the environment of secondary minerals: actinolite, and/or albite, prehnite, pumpellyite (rarely calcite) (Figure 7). The composition of the studied spinels is similar, and they are characterized by high contents of $Al_2O_3$, $Cr_2O_3$, and FeOtot (Table 6). The minor differences in spinel composition are presumably caused by the mineral environment in which they occur. Overall, we can divide the studied spinels according to the type of surrounding minerals into: (1) spinels in actinolites and (2) spinels in associations—albite, prehnite, and chlorite. Compared to the second group, the spinels in the first group (in actinolites) have a higher Fe (and Zn) content and lower Al, Cr, and Mg (less Ti). The most significant differences are in the contents of Mg and $Fe^{2+}$ (Figure 8A). In the figure, we can also observe a significant negative correlation between these elements. The composition of spinels corresponds to spinels from volcanic rocks [50,51], so they are not relics of spinels from upper mantle (Figure 8B).

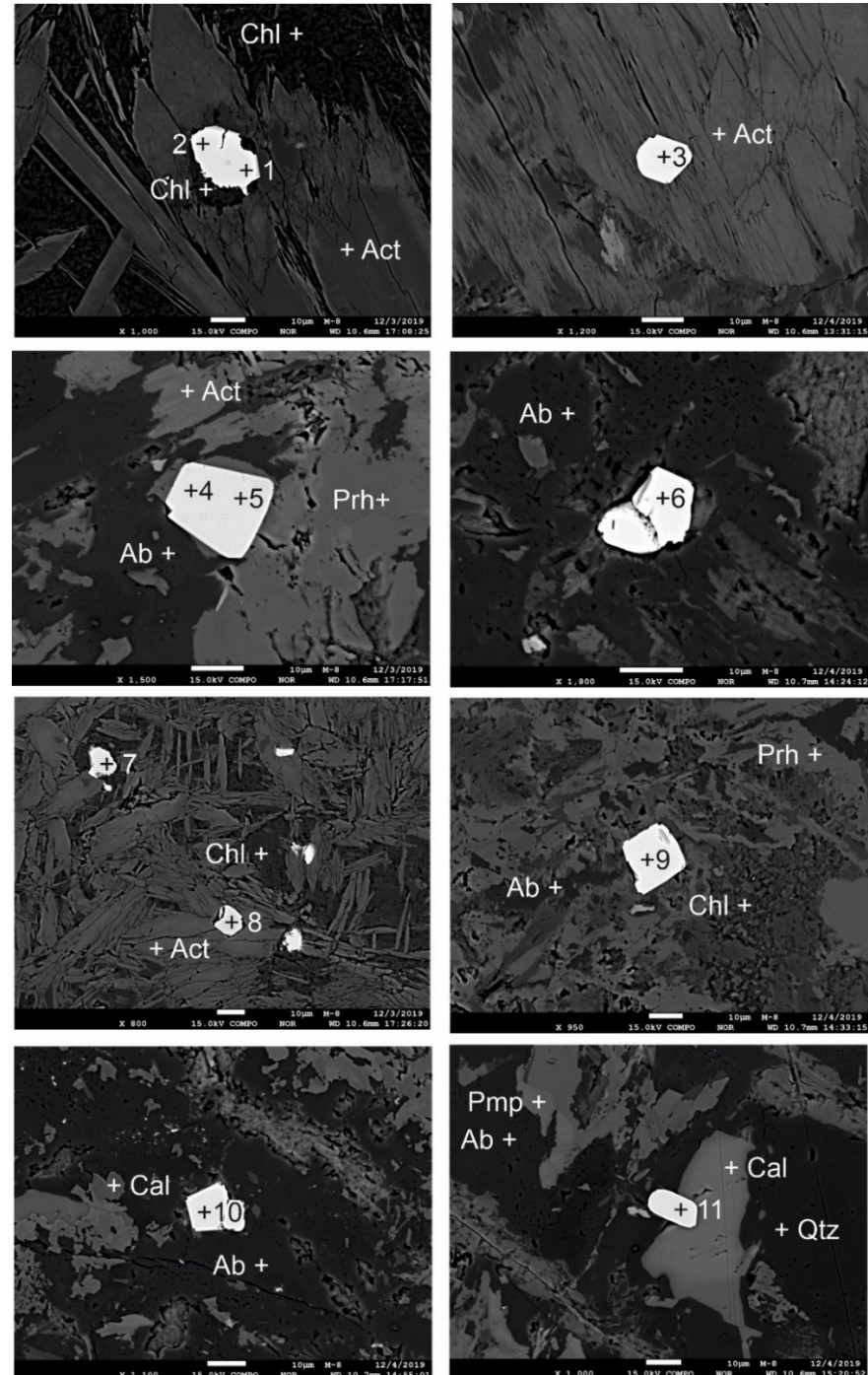

**Figure 7.** Back scattered electron (BSE) images of basalts from Malužiná; Act = actinolite, Chl = chlorite, Ab = albite, Prh = prehnite, Pmp = pumpellyite, Cal = calcite, Spl = spinels; the numbers in figures correspond to those in Tables 2–5.

**Table 6.** Selected analyses of spinels.

| Anal. N. | 1 | 2 | 3 | 4 | 5 | 6 | 7 | 8 | 9 | 10 | 11 |
|---|---|---|---|---|---|---|---|---|---|---|---|
| $SiO_2$ | 0.09 | 0.05 | 0.11 | 0.08 | 0.07 | 0.138 | 0.11 | 0.08 | 0.00 | 0.05 | 0.08 |
| $TiO_2$ | 1.11 | 0.86 | 1.53 | 1.93 | 1.93 | 1.322 | 0.83 | 0.66 | 2.31 | 1.86 | 1.85 |
| $Al_2O_3$ | 12.76 | 13.19 | 9.29 | 20.71 | 20.39 | 16.265 | 16.55 | 15.33 | 17.47 | 20.28 | 20.42 |
| $Cr_2O_3$ | 27.59 | 27.81 | 30.24 | 32.86 | 33.70 | 36.020 | 29.00 | 29.35 | 34.45 | 33.68 | 32.66 |
| FeO | 26.98 | 26.47 | 27.43 | 17.91 | 18.25 | 13.062 | 25.98 | 25.70 | 18.46 | 17.44 | 17.08 |
| $Fe_2O_3$ | 25.97 | 25.26 | 26.17 | 14.14 | 13.42 | 21.435 | 20.60 | 21.84 | 15.26 | 14.06 | 14.11 |
| MnO | 0.41 | 0.46 | 0.45 | 0.34 | 0.34 | 0.380 | 0.39 | 0.40 | 0.30 | 0.32 | 0.37 |
| MgO | 3.69 | 3.87 | 2.95 | 10.67 | 10.44 | 8.556 | 4.63 | 4.71 | 9.89 | 10.93 | 10.91 |
| ZnO | 0.31 | 0.33 | 0.19 | 0.14 | 0.01 | 0.10 | 0.46 | 0.31 | 0.12 | 0.14 | 0.16 |
| $V_2O_5$ | 0.20 | 0.24 | 0.34 | 0.29 | 0.26 | 0.23 | 0.23 | 0.19 | 0.31 | 0.22 | 0.23 |
| NiO | 0.02 | 0.11 | 0.05 | 0.00 | 0.02 | 0.00 | 0.06 | 0.08 | 0.01 | 0.04 | 0.01 |
| total | 99.13 | 98.62 | 98.75 | 99.06 | 98.83 | 97.51 | 98.83 | 98.65 | 98.58 | 99.01 | 97.89 |
| Formula based on 4 oxygens | | | | | | | | | | | |
| Si | 0.003 | 0.002 | 0.004 | 0.003 | 0.002 | 0.005 | 0.004 | 0.003 | 0.000 | 0.002 | 0.002 |
| Ti | 0.029 | 0.023 | 0.041 | 0.047 | 0.047 | 0.033 | 0.022 | 0.017 | 0.058 | 0.045 | 0.045 |
| Al | 0.526 | 0.545 | 0.393 | 0.795 | 0.779 | 0.642 | 0.674 | 0.622 | 0.681 | 0.772 | 0.785 |
| Cr | 0.763 | 0.771 | 0.859 | 0.847 | 0.864 | 0.953 | 0.793 | 0.799 | 0.901 | 0.860 | 0.842 |
| $Fe^{3+}$ | 0.681 | 0.663 | 0.704 | 0.339 | 0.324 | 0.329 | 0.527 | 0.563 | 0.375 | 0.338 | 0.342 |
| $Fe^{2+}$ | 0.786 | 0.772 | 0.820 | 0.478 | 0.489 | 0.600 | 0.739 | 0.736 | 0.504 | 0.466 | 0.460 |
| Mn | 0.012 | 0.014 | 0.014 | 0.009 | 0.009 | 0.011 | 0.011 | 0.012 | 0.008 | 0.009 | 0.010 |
| Mg | 0.193 | 0.202 | 0.158 | 0.518 | 0.504 | 0.427 | 0.239 | 0.242 | 0.488 | 0.526 | 0.530 |

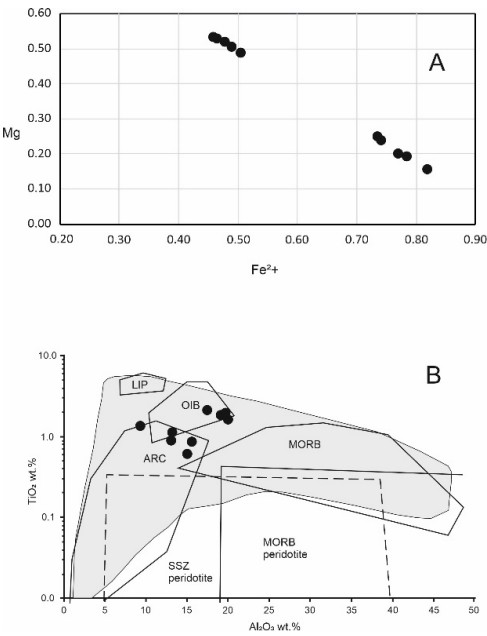

**Figure 8.** (**A**) Mg vs. $Fe^{2+}$ plot; (**B**) $TiO_2$ versus $Al_2O_3$ plot for studied spinels. Large Igneous Province (LIP), Ocean Island Basalts (OIB), Island Arc magmas (ARC), Mid Ocean Ridge Basalts (MORB), volcanic rocks (grey field), Suprasubduction Zone (SSZ), MORB and peridotite fields after Lenaz et al. [54] and Kamenetsky et al. [55].

## 5. Geochemistry

Seven samples were studied for chemical composition (Table 7). $SiO_2$ content in the samples varied from 44 to 54 wt.%. The mg# number was low to moderate, ranging from 25 to 68. The contents of $TiO_2$ correspond accordingly (1.2–2.9 wt.%). Based on TAS classification, the rock studied corresponds with the basalt and basaltic andesite field of sub-alkaline rocks (Figure 9A). However, strong secondary alteration increased alkali contents and shifted sample composition within the field of alkaline rocks. As alkali are highly mobile during super-imposed hydrothermal processes and regional metamorphism, the high field strength elements (HFSE) Zr, Nb, Y, and $TiO_2$ were used for rock classification. Based on Winchester and Floyd [56] Nb/Y vs. $Zr/TiO_2$ discrimination diagram, the analysed rocks form a coherent group within the andesite/basalt field (Figure 9B). For a more precise geotectonic classification of the studied rocks, different ternary diagrams were used (Figure 10A [57]; Figure 10B [58]). In both diagrams, the analyses of the studied rocks fall within the field of calc-alkali, and/or continental basalts. The REE normalized curve (Figure 11A [59]) shows a slight enrichment in light REE, with no significant HREE fractionation. The REE pattern is slightly different in M-3 sample, with the lowest LREE contents; this may be due to a different mineral composition, which is reflected in a significantly lower CaO content compared to other samples. Nevertheless, negative Eu anomalies are slight to negligible for all the samples (0.72–0.97). In the primitive mantle normalized plots (Figure 11B), the analysed samples show a marked depletion of Rb, Ba, Nb, Ta, and Sr, as well as a slight depletion of Ti and HREE. On the other hand, the enrichment in Th, U, accompanied also by a slight enrichment of P, Nd, and Zr, is observed.

**Table 7.** Chemical composition of studied samples.

| Sample | M-1 | M-1A | M-2 | M-3 | M-4 | M-5 | M-7 |
|---|---|---|---|---|---|---|---|
| $SiO_2$ | 50.24 | 49.14 | 50.54 | 49.57 | 44.52 | 53.92 | 53.42 |
| $TiO_2$ | 1.42 | 2.09 | 2.14 | 1.61 | 1.26 | 1.20 | 2.46 |
| $Al_2O_3$ | 15.87 | 16.80 | 15.98 | 17.88 | 17.26 | 15.24 | 16.72 |
| $Fe_2O_3$* | 7.12 | 10.88 | 10.21 | 9.95 | 6.37 | 9.65 | 11.36 |
| $Cr_2O_3$ | 0.02 | 0.01 | <0.002 | 0.01 | 0.02 | 0.01 | <0.002 |
| CaO | 8.33 | 7.58 | 5.88 | 1.12 | 8.31 | 4.55 | 0.69 |
| MnO | 0.12 | 0.17 | 0.13 | 0.09 | 0.10 | 0.09 | 0.04 |
| MgO | 5.46 | 5.61 | 1.68 | 9.07 | 6.07 | 3.40 | 5.77 |
| $Na_2O$ | 4.21 | 3.53 | 7.31 | 4.45 | 2.83 | 6.66 | 4.79 |
| $K_2O$ | 0.88 | 1.35 | 0.27 | 0.46 | 1.63 | 0.14 | 0.17 |
| $P_2O_5$ | 0.24 | 0.32 | 0.49 | 0.18 | 0.20 | 0.21 | 0.43 |
| LOl | 5.90 | 2.20 | 5.20 | 5.40 | 11.20 | 4.80 | 3.90 |
| Total | 99.81 | 99.68 | 99.83 | 99.79 | 99.77 | 99.87 | 99.75 |
| TOT/C | 0.74 | 0.03 | 1.11 | 0.09 | 1.77 | 0.69 | <0.02 |
| TOT/S | <0.02 | 0.02 | <0.02 | <0.02 | <0.02 | <0.02 | <0.02 |
| Sc | 30.00 | 25.00 | 20.00 | 26.00 | 23.00 | 18.00 | 25.00 |
| Ba | 152.00 | 697.00 | 87.00 | 51.00 | 206.00 | 48.00 | 439.00 |
| Be | 2.00 | 2.00 | 1.00 | <1 | <1 | <1 | 2.00 |
| Co | 28.20 | 36.8 | 21.10 | 35.00 | 34.40 | 24.50 | 26.80 |
| Cs | 2.30 | 2.90 | 0.40 | 2.50 | 5.10 | 0.40 | 0.70 |
| Ga | 16.90 | 19.20 | 19.20 | 17.40 | 15.60 | 14.50 | 20.00 |
| Hf | 5.20 | 5.40 | 6.80 | 3.90 | 3.90 | 4.30 | 6.30 |

**Table 7.** *Cont.*

| Sample | M-1 | M-1A | M-2 | M-3 | M-4 | M-5 | M-7 |
|--------|------|------|------|------|------|------|------|
| Nb | 9.78 | 9.70 | 8.60 | 3.50 | 7.40 | 6.30 | 11.10 |
| Rb | 35.10 | 43.70 | 8.20 | 11.30 | 59.10 | 4.40 | 5.40 |
| Sn | 1.00 | 2.00 | 2.00 | 1.00 | 1.00 | 2.00 | 2.00 |
| Sr | 253.10 | 375.50 | 108.00 | 133.60 | 59.80 | 224.80 | 57.90 |
| Ta | 0.60 | 0.60 | 0.70 | 0.30 | 0.50 | 0.40 | 0.80 |
| Th | 5.30 | 3.00 | 3.30 | 1.60 | 3.90 | 3.30 | 6.40 |
| U | 1.20 | 1.00 | 1.20 | 0.40 | 1.10 | 0.70 | 2.80 |
| V | 187.00 | 218.00 | 137.00 | 200.00 | 170.00 | 166.00 | 262.00 |
| W | 6.60 | 6.00 | 7.60 | 4.20 | 2.30 | 5.60 | 11.90 |
| Zr | 200.20 | 224.10 | 271.80 | 149.00 | 153.40 | 147.50 | 231.60 |
| Y | 33.60 | 39.60 | 53.20 | 24.40 | 26.00 | 23.60 | 45.30 |
| La | 23.80 | 17.30 | 17.70 | 8.10 | 21.00 | 15.70 | 19.90 |
| Ce | 53.50 | 41.60 | 44.70 | 21.30 | 47.30 | 38.90 | 50.20 |
| Pr | 6.69 | 5.73 | 6.36 | 2.95 | 5.82 | 5.22 | 7.04 |
| Nd | 27.90 | 25.70 | 29.70 | 13.10 | 24.60 | 21.20 | 30.90 |
| Sm | 6.20 | 6.46 | 7.80 | 3.54 | 4.99 | 4.51 | 7.30 |
| Eu | 1.72 | 1.88 | 2.37 | 0.91 | 1.63 | 1.43 | 2.05 |
| Gd | 6.45 | 7.19 | 9.56 | 4.19 | 5.01 | 4.74 | 8.40 |
| Tb | 0.99 | 1.11 | 1.49 | 0.69 | 0.76 | 0.71 | 1.33 |
| Dy | 6.11 | 7.07 | 9.35 | 4.48 | 4.57 | 4.46 | 8.16 |
| Ho | 1.17 | 1.43 | 1.88 | 0.93 | 0.91 | 0.99 | 1.67 |
| Er | 3.36 | 4.05 | 5.50 | 2.73 | 2.61 | 2.61 | 4.76 |
| Tm | 0.51 | 0.60 | 0.79 | 0.42 | 0.37 | 0.40 | 0.70 |
| Yb | 3.19 | 3.77 | 5.05 | 2.59 | 2.36 | 2.48 | 4.31 |
| Lu | 0.50 | 0.58 | 0.78 | 0.42 | 0.35 | 0.40 | 0.68 |
| Ni | 41.00 | 50.00 | <20 | 32.00 | 81.00 | 35.00 | <20 |

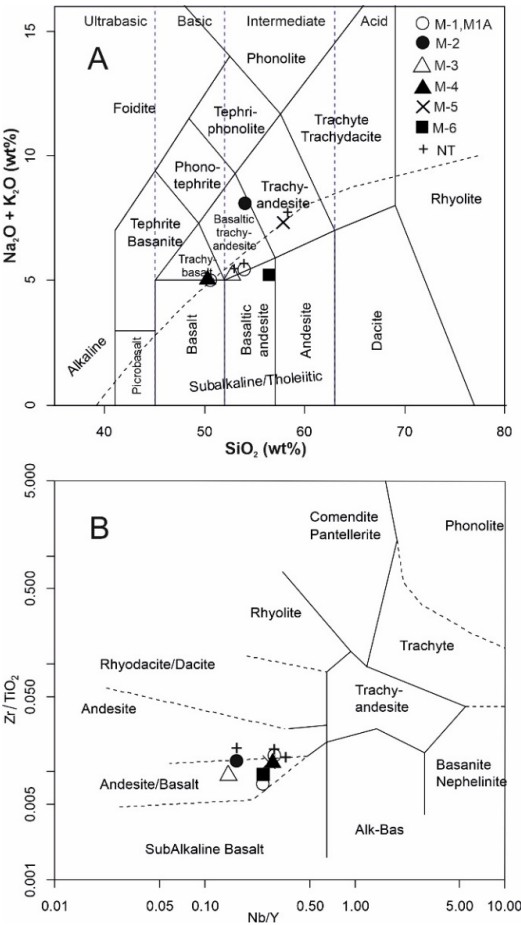

**Figure 9.** (**A**) TAS plot [60]; (**B**) Nb/Y vs. Zr/TiO$_2$ plot for studied basalts [56]. Sample M-1—M7 from Table 7, NT—analyses of paleobasalts from Vozár et al. [61].

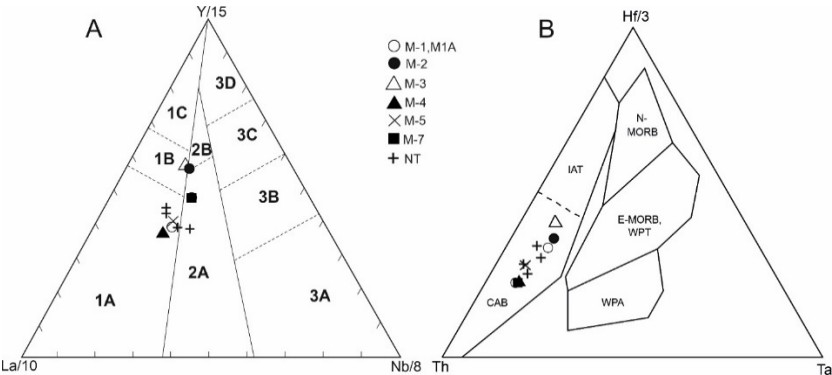

**Figure 10.** (**A**) The La-Y-Nb: discrimination diagram for basalts [58] Field 1—volcanic arc basalts, field 2—continental basalts and field 3—oceanic basalts. The subdivision of the fields as follows: 1A = calc-alkaline basalts, 1C = volcanic arc tholeiites, 1B = overlap is between 1A and 1C, 2A = continental basalts, 2B = back-are basin basalts, 3A = alkaline basalts from intercontinental rift, 3B, 3C = E-type MORB (enriched MORB), 3D = N type– MORB; B) The Th-Hf-Ta: discrimination diagram for basalts [57]. (**B**) The field are defined as follows: N-MORB (normal Mid Ocean Ridge Basalt), E-MORB and WPT (within-plate tholeiites), WPA (within-plate alkali basalts), IAT (Island arc tholeiites, Hf/Th > 3.0) and CAB (calc-alkali basalts; Hf/Th < 3.0). The broken lines indicate transitional zones between basalt types. Sample M-1—M7 from Table 7, NT—analyses of paleobasalts from Vozár et al. [61].

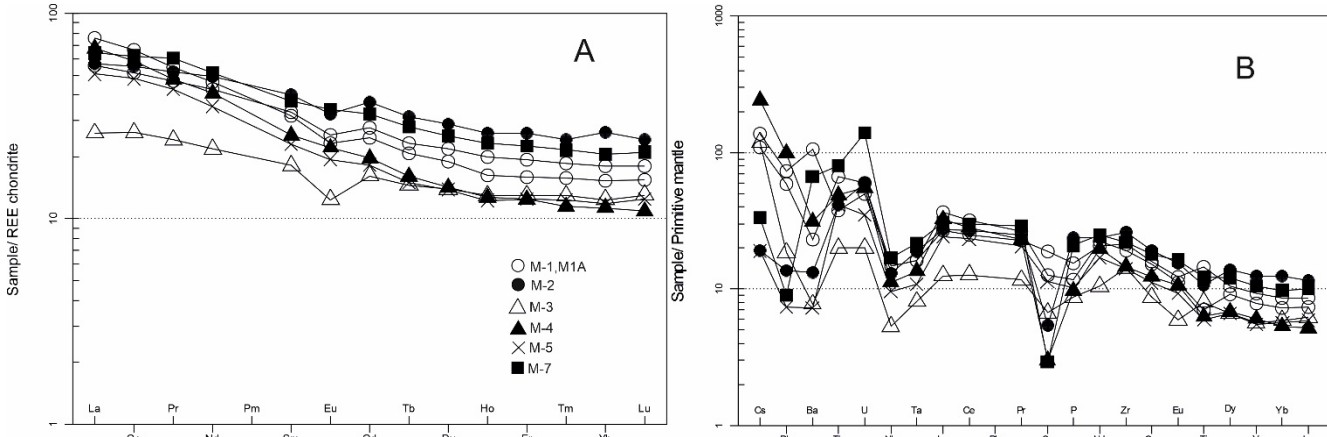

**Figure 11.** (**A**) REE concentration normalized to the chondrite composition [59]; (**B**) Trace element concentrations normalized to the composition of the primitive mantle [62]. Sample M-1—M7 from Table 7.

The Sr, Nd, and Pb isotopic compositions were determined in five samples (Table 8, whole rocks) and together with the three previously published results [61] the whole set is represented by eight samples. Two samples from the diorite dykes and three samples from the first and second eruption phases were analysed. The Nd-Sr isotope signature of the studied basalts are scattered around the value of $^{143}$Nd/$^{144}$Nd for CHUR, where $^{143}$Nd$^{144}$Nd ratios are relatively stable and $^{87}$Sr/$^{86}$Sr values are relatively varied. (Figure 12). Some of the points lie in field IV, and/or its vicinity, indicating a relatively high proportion of crustal contamination in the formation of these basalts. The study of Pb isotopes gives us similar conclusions. The diagram (Figure 13) illustrates the isotopic ratio of $^{207}$Pb/$^{204}$Pb vs. $^{206}$Pb/$^{204}$Pb. The basalt analyses lie in the field of EMII, and/or they partially overlap with the Saxo-Thuringian upper crust fields. Compared to the MORB field, they have higher $^{207}$Pb$^{204}$Pb. The EMII field is defined as a mantle source enriched with crustal materials [63,64]. Crustal materials can be incorporated into mantle material in different ways [65,66]. This position indicates a relatively high degree of crustal contamination. This is especially evident from $^{206}$Pb/$^{204}$Pb isotope ratios, which correspond to values around 19. This Pb isotope ratio is considered to be the most sensitive indicator of crustal contamination.

**Table 8.** Sr, Nd, and Pb isotope analytical data from the studied samples.

| Sample | M-1 | M-3 | M-4 | M-5 | M-7 |
|---|---|---|---|---|---|
| $^{206}$Pb/$^{204}$Pb | 18.92209939 | 18.7368185 | 19.03282702 | 18.57438457 | 19.14024785 |
| ±2s+ | 0.011112606 | 0.01491399 | 0.012461895 | 0.010005923 | 0.019462799 |
| $^{207}$Pb/$^{204}$Pb | 15.66053538 | 15.6515818 | 15.66947086 | 15.6394058 | 15.6842099 |
| ±2s+ | 0.010983742 | 0.014073549 | 0.011949346 | 0.010340995 | 0.017133993 |
| $^{208}$Pb/$^{204}$Pb | 39.09866217 | 38.74599643 | 39.10893752 | 38.5879748 | 39.07321516 |
| ±2s+ | 0.032498962 | 0.038868704 | 0.034110976 | 0.030599558 | 0.045900435 |
| $^{207}$Pb/$^{206}$Pb | 0.827633773 | 0.83534005 | 0.823288311 | 0.841989613 | 0.819437825 |
| ±2s+ | 0.000174809 | 0.000224997 | 0.000184886 | 0.000177841 | 0.000195686 |
| $^{208}$Pb/$^{206}$Pb | 2.066304938 | 2.067915649 | 2.054823635 | 2.077492136 | 2.04142499 |
| ±2s+ | 0.000776447 | 0.000847523 | 0.000732947 | 0.000759999 | 0.000767098 |
| $^{87}$Sr/$^{86}$Sr | 0.707907300 | 0.706442300 | 0.713264800 | 0.706534300 | 0.708874500 |
| +/−abs | $4.95535 \times 10^{-6}$ | $6.35798 \times 10^{-6}$ | $4.27959 \times 10^{-6}$ | $4.23921 \times 10^{-6}$ | $4.96212 \times 10^{-6}$ |
| $^{143}$Nd/$^{144}$Nd | 0.512546651 | 0.512868216 | 0.51259686 | 0.512701087 | 0.512593255 |

**Table 8.** *Cont.*

| Sample | M-1 | M-3 | M-4 | M-5 | M-7 |
|---|---|---|---|---|---|
| +/−abs | $5.63801 \times 10^{-6}$ | $6.15442 \times 10^{-6}$ | $5.63857 \times 10^{-6}$ | $5.63971 \times 10^{-6}$ | $3.58815 \times 10^{-6}$ |
| Age (Ma) | 265 | 265 | 265 | 265 | 265 |
| $\varepsilon$Nd | 0.33 | 5.62 | 1.7 | 3.54 | 0.95 |

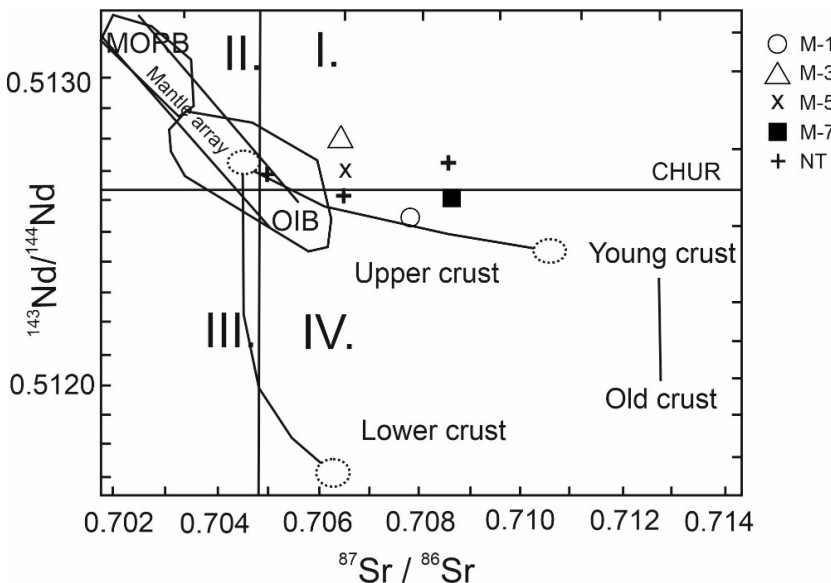

**Figure 12.** $^{143}$Nd/$^{144}$Nd vs. $^{87}$Sr/$^{86}$Sr discriminant diagram, CHUR—chondrite uniform reservoir; OIB—oceanic island basalt; MORB—mid ocean ride basalts, according to Faure [67]. Sample M-1—M7 from Table 8, NT—analyses of paleobasalts from Vozár et al. [61].

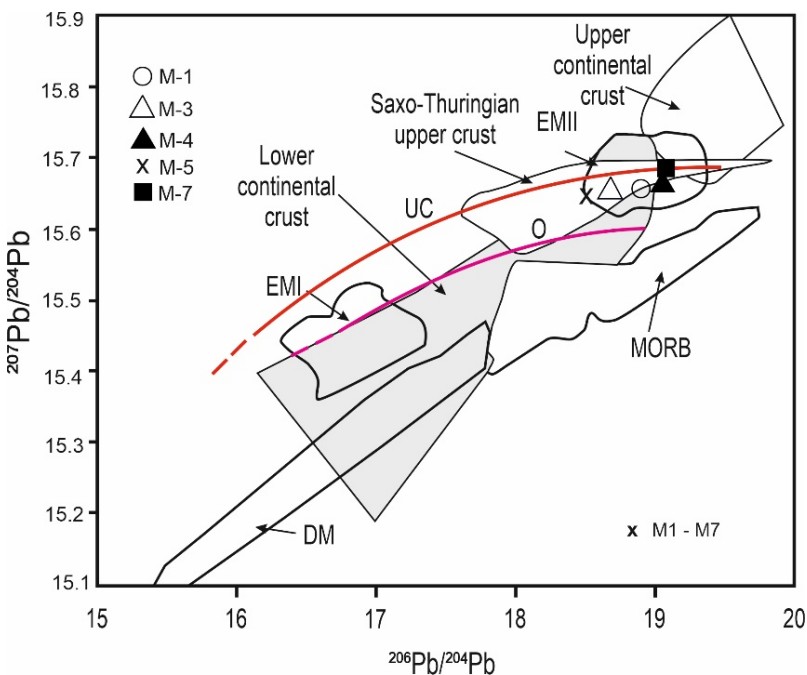

**Figure 13.** 207Pb/204Pb vs. 206Pb/204Pb in studied basalt; Saxo-Thuringian upper crust are taken from Romer and Hahne [68] and DM, EMI, EMII (from Zindler and Hart [63]).Orogene (O) and upper crust (UC) Pb evolution curves are after Zartman and Doe [69].Other fields from Sommer et al. [70]; Sample M-1—M7 from Table 8.

## 6. Discussion and Interpretation

From the geochemical point of view, the studied Hronicum unit basic rocks can be classified as alkaline–calcium basalts with an affinity for continental basalts. (Figure 10A,B). We used various immobile elements and their ratios to determine the genesis of these rocks (Figure 14). The diagrams show that the rocks studied are not a product of fractional crystallization or a combination of assimilation and fractional crystallization. Their genesis is relatively complex, and it is probably a matter of mixing (contamination) of mantle sources with crust material. If we estimate the proportion of upper crust material and the primary source of N-MORB [71], the studied basalts appear to contain approximately 10% of crust material.

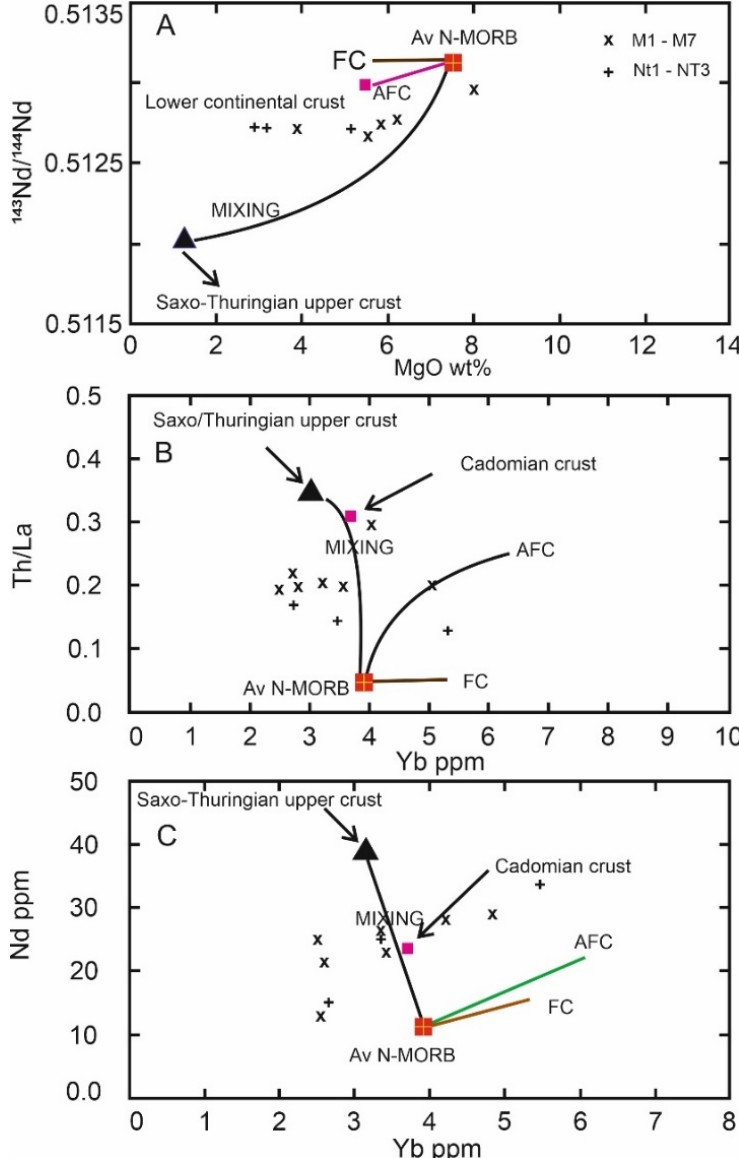

**Figure 14.** (**A**) 143Nd/144Nd vs. MgO; (**B**) Th/La vs. Yb; (**C**) Nd vs. Yb plots showing the results of FC, AFC, and mixing modeling results. N-MORB average is from Hofmann [72] and Ito et al. [73]. Saxo-Thuringian upper crust from Romer and Hahne [68], average lower continental crust from Taylor and McLennan [74] and Cadomian crust from Linnemann and Romer [75]. AFC—assimilation with fractional crystallization, FC—fractional crystallization. Others from Abdelfadil et al. [76]. Sample M-1—M7 from Tables 7 and 8, NT—analyses of paleobasalts from Vozár et al. [61].

However, the chemical and isotopic compositions of 1st and 2nd eruption phase volcanic rocks from the Malužina Formation do not show greater differences, despite their different stratigraphic position. Perhaps the only difference is the slightly higher value of $^{87}Sr/^{86}Sr$ ratio in the volcanics of the 1st eruption phase (samples M-7, M-4, and NT-2; Table 8), which indicates a relatively high degree of crustal contamination, compared to the 2nd eruption phase. The $^{87}Sr/^{86}Sr$ ratio in gabbro-diorite dykes is similar to the volcanics of the 1st eruption phase (samples M-1A and NT-1; Table 8), which could indicate their concurrence. This is also indicated by a slightly higher LILE and LREE enrichment in the volcanics of the 1st eruption phase compared to the 2nd eruption phase (Figure 10A,B). Conversely, $^{143}Nd/^{144}Nd$ isotope ratio increases slightly in the volcanics of the 2nd eruption phase compared to the volcanics of the 1st eruption phase, which may also correspond to the differences in the degree of contamination by crustal material. Similarly, variations in $^{206}Pb/^{204}Pb$ isotope ratio indicate higher values (19.14–19.03) in 1st eruption phase volcanics compared to 2nd eruption phase volcanics (18.37–18.73; Table 8). This trend is probably a result of upper crust contamination, as evidenced by higher $^{206}Pb/^{204}Pb$ isotope ratios consistent with the increasing $^{87}Sr/^{86}Sr$ isotope ratios. A similar trend is shown, for example, by Paraná basalts [77,78].

The values of $^{206}Pb/^{204}Pb$ (18.57–19.14), $^{207}Pb/^{204}Pb$ (15.63–15.68) and $^{208}Pb/^{204}Pb$ (38.58–39.10) are significantly higher in the volcanics of the Malužina Formation compared to depleted mantle (DM) line—MORB basalts, but overall they do not show a large dispersion. We must therefore assume a single magmatic source for both volcanic phases and the dyke complex. As already mentioned, the isotopic composition indicates an affinity with the EM II mantle source. However, the values of La/Sm and Sm/Yb (4.2–2.3 and 1.4–2.1, respectively) are low, indicating derivation from primary mantle and no significant mantle or crust contamination with lithospheric mantle [79]. This would indicate a rapid ascent of magma, which was a reason why it was not contaminated to a greater extent with crust material.

Basic dykes appear in all occurrences of the Hronicum within a dark gray formation referred to as the Nižná Boca Formation. Based on the remains of macroflora [24] and also microflora [25], the Nižná Boca Formation was included in the Stephanian (according to the Regional Stratigraphic Scale of Central Europe in Vozárová and Vozár [3]), and later correlated with the uppermost Pennsylvanian (Lopingian) according to ICS chart (2019) of the stratigraphic scale. Logically, this led to the interpretation that the basic dykes are of pre-Permian age. However, the analysis of detrital zircons [8] showed that the upper age limit of the Nižná Boca Formation should be shifted as far as the Lower Cisuralian (Asselian-Sakmarian). On the other hand, the results of U-Pb LA-ICP-MS dating of apatite point to a younger age of 254 ± 23 Ma (Figure 15), which (although with a relatively large scatter of values) is evidence of their Permian age. There are more possibilities of interpretation:

- The basic dykes intruded together with the later, second volcanic phase in the Permian, at the time of maximum extension. The age of the second eruption phase in the Malužina Formation is indicated on the basis of the microflora as Thuringien (Planderová [25] according to the regional Central European scale), later correlated with the standard ICS stratigraphic scale as the Wuchiapingian-Changhsingian (period after the Capitanian degree; ca. 265 Ma), when Pangea breakup began in general and roughly coincides with the Illawara Magnetic Horizon [31,80]. This period is also associated with the so-called "Mid-Permian Episode" by Deroin and Bonin [81], associated with transform strike-slip movement and the onset of maximum extension, leading to Pangea breakage.
- However, it is quite likely that the basic dykes do not have the same age. The extension in the Hronicum sedimentation space took place in several stages and intrusions of dykes could be associated with each stage of extension.
- Another possibility is that during the extrusion period of the second eruption phase, the elements were remobilized and completely overheated, which resulted in recrys-

tallization of older apatites or rather crystallization of new ones either in basic dykes or in the volcanics of the first eruption phase. This could explain the large variance of ages in dating, but at the same time it would be consistent with the Upper Permian age (Lopingian) of the second eruption phase.

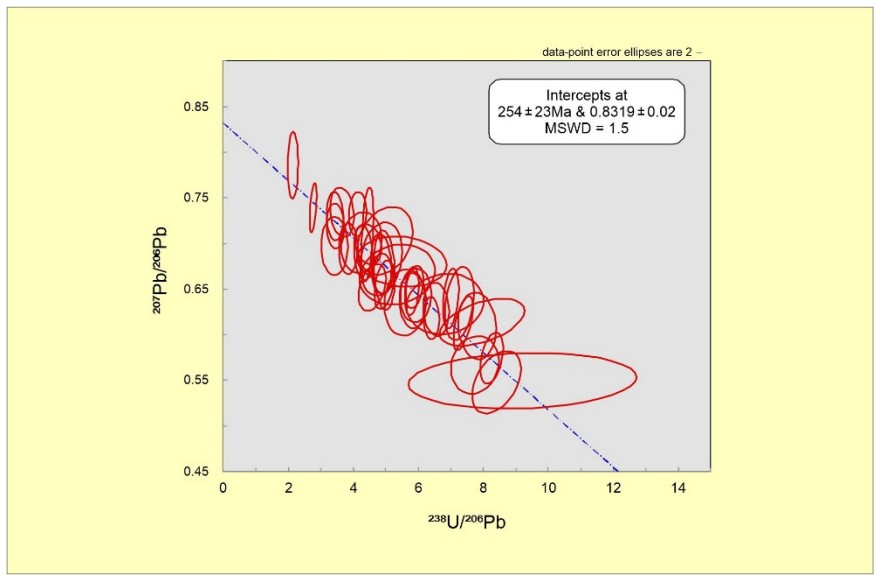

**Figure 15.** The LA ICP MS analysis of apatites from the Nižná Boca locality; sample M-8.

## 7. Conclusions

1. The basalts of the Malužina Formation are strongly altered; clinopyroxene, plagioclase, and amphibole are locally preserved from the primary minerals. In basalts the predominant textures are intersertal, pilotaxitic, hyalopilitic. Amygdaloid and vesicular structures have been commonly identified.

2. Composition of main rock-forming minerals in different localities is similar. The clinopyroxenes are relatively homogenous in chemical composition, with only rarely observed darker zones in larger grains (Figure 2e) rich in Mg, and/or depleted of Fe. Based on the classification of pyroxenes these Cpx correspond to augite. There are two types of amphibole; the older amphibole coexisted with Cpx and has a composition corresponding to pargasite. The younger amphibole has a composition corresponding to actinolite/tremolite. Biotite is strongly altered and the biotites studied lie in the field of calc-alkaline rocks.

3. Rare small grains of Cr-spinel have been observed in the gabbro-diorite dyke. Spinel grains probably formed part of the primary mineral association (clinopyroxenes, plagioclases, pargasite). The composition of spinels corresponds to spinels from volcanic rocks.

4. From the geochemical point of view, the studied rocks can be classified as alkaline–calcium basalts with affinity for continental basalts. The basalts of the Malužina Formation show a significant depletion of Nb and Ta compared to La and Th when normalized to the primitive mantle. They are also depleted of P, which corresponds to the very low content of apatite in the studied volcanics. These features and the very low content of primary amphibole or apatite in the studied volcanics rather indicate a "dry" magmatic source.

5. Based on the distribution trace elements and Sr-Nd-Pb isotope composition a magmatic source of EM II type can be assumed. The isotopic composition, depletion of Nb and Ta, and limited variability in the ratios of incompatible elements, indicate only weak crustal assimilation, which could be explained by a rapid ascent of magma to the surface.

6. The two eruption phases correspond to two significant extension pulses during the development of the Hronicum original sedimentation basin in Permian. The presumed magmatic source of both eruption phases was identical. The differences indicated by the distribution of REE, incompatible elements, as well as Sr and Pb isotopes, were due to the relatively higher extents of mantle magma contamination by crust material in the volcanics of the first eruption phase.

**Author Contributions:** Conceptualization, J.S., A.V. and V.Š.; methodology, J.S., A.V. and J.B.; software, J.B.; validation, J.S. and A.V.; formal analysis, J.V. and Š.F.; investigation and resources, J.S., A.V., J.V., Š.F., V.Š. and J.B.; data curation, J.B.; writing—original draft preparation, J.S. and A.V.; writing—review and editing, J.S. and V.Š.; visualization and supervision, J.S.; project administration, J.S. and V.Š. All authors have read and agreed to the published version of the manuscript.

**Funding:** Informed consent was obtained from all subjects involved in the study.

**Institutional Review Board Statement:** Not applicable.

**Informed Consent Statement:** Not applicable.

**Data Availability Statement:** Not applicable.

**Acknowledgments:** This research was supported by grants VEGA 2/0006/19, VEGA 1/0237/18, 033UMB-4/2021 and APVV 19-0065, APVV-0146-16.

**Conflicts of Interest:** We confirm that the authors have declare no conflict of interest.

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
