# Peer review of "Implication of Mineralogy and Isotope Data on the Origin of the Permian Basic Volcanic Rocks of the Hronicum (Slovakia, Western Carpathians)"

_minerals, doi:10.3390/min11080841_

Round 1

Reviewer 1 Report

The manuscript can be accepted in its current form.
We only suggest a few minor corrections to the figure captions

Author Response

All suggestions for changes in the text and description of the images have been incorporated. Thank you.

Reviewer 2 Report

- The conclusions of the paper could be further improved.

- The bibliography is dense. Some of the references are not appropriate (abstracts or irrelevant publications) (e.g. [12], [26], [27], [30], [33], [71]).

Author Response

  • We have made improvements to the text of the chapter and conclusion reflecting on your proposed comments and suggestions. 
  • After the review by all co-authors we have concluded that all references you pointed out in your review were either book chapters or manuscripts. We consider them relevant for the current submission.

Thank you.

Reviewer 3 Report

The paper evaluates mineralogy and isotope data of the Permian basic volcanic rocks in Slovakia to provide insight into its origin and the significance. The study is interesting and reports numerous results obtained by modern instrumental techniques. Therefore, there is a need to publish it. Nevertheless, as there are still some issues that are to be addressed, my overall recommendation is Minor revision prior to its publication. Specific comments and suggestions are given below.

I suggest revising sections headlines. Section 1.1. Geological background (now within section 1. Introduction) could be a part of newly created section 2. Samples and methods (2.1. Geological background and samples collection (e.g.), and, 2.2. Analytical methods). Then, it should be clear that the results and measured data are presented – i.e., section 3. should be Results (or something like this). In this section, current section 3. Petrography and Mineralogy should be presented as 3.1., but as being now too long (10 pages) it could be divided into subsections. Section 4. Geochemistry could be now section 3.2. Then, section 5. Discussion and interpretation will be section 4, and Conclusions will be section 5.

References in the text – please use [3-5] instead of [3,4,5] etc.

Please check the manuscript for proper writing of subscripts (lower index) in chemical formulas TiO2, Al2O3 etc.

Why Fe2+ and Mn2+ are used twice in Tab. 3? It could be explained below the table.

Please add “+” to Fe2 or Fe3 in Tabs. 2 and 6.

Fig. 8 – there is no need to use more than 1 decimal on both axes (as there are all 0)

Author Response

  • Chapter numbering has been simplified. 
  • Suggested changes to the format of references in the text has been adjusted. 
  • We have unified the format of the subscripts and superscripts in the chemical formulas in the text as well as in the tables. 
  • We have adjusted all requested decimal places and table format as requested.

Thank you for your recommendation.